# Microcephaly-associated protein WDR62 shuttles from the Golgi apparatus to the spindle poles in human neural progenitors

**Claudia Dell'Amico[1], Marilyn M Angulo Salavarria[1], Yutaka Takeo[2], Ichiko Saotome[2], Maria Teresa Dell'Anno[3], Maura Galimberti[4,5], Enrica Pellegrino[1,6], Elena Cattaneo[4,5], Angeliki Louvi[2]\*, Marco Onorati[1]\***

[1]Department of Biology, Unit of Cell and Developmental Biology, University of Pisa, Pisa, Italy; [2]Departments of Neurosurgery and Neuroscience, Yale School of Medicine, New Haven, United States; [3]Fondazione Pisana per la Scienza ONLUS, San Giuliano Terme, Italy; [4]Dipartimento di Bioscienze, Università degli Studi di Milano, Milan, Italy; [5]INGM, Istituto Nazionale Genetica Molecolare, Milan, Italy; [6]Host-Pathogen Interactions in Tuberculosis Laboratory, The Francis Crick Institute, London, United Kingdom

**\*For correspondence:**
angeliki.louvi@yale.edu (AL);
marco.onorati@unipi.it (MO)

**Competing interest:** The authors declare that no competing interests exist.

**Abstract** WDR62 is a spindle pole-associated scaffold protein with pleiotropic functions. Recessive mutations in *WDR62* cause structural brain abnormalities and account for the second most common cause of autosomal recessive primary microcephaly (MCPH), indicating WDR62 as a critical hub for human brain development. Here, we investigated WDR62 function in corticogenesis through the analysis of a C-terminal truncating mutation (D955AfsX112). Using induced Pluripotent Stem Cells (iPSCs) obtained from a patient and his unaffected parent, as well as isogenic corrected lines, we generated 2D and 3D models of human neurodevelopment, including neuroepithelial stem cells, cerebro-cortical progenitors, terminally differentiated neurons, and cerebral organoids. We report that WDR62 localizes to the Golgi apparatus during interphase in cultured cells and human fetal brain tissue, and translocates to the mitotic spindle poles in a microtubule-dependent manner. Moreover, we demonstrate that WDR62 dysfunction impairs mitotic progression and results in alterations of the neurogenic trajectories of iPSC neuroderivatives. In summary, impairment of WDR62 localization and function results in severe neurodevelopmental abnormalities, thus delineating new mechanisms in the etiology of MCPH.

## Editor's evaluation

This paper is of interest to neurobiologists studying brain development and potentially to cell biologists in general. The study utilizes 2D and 3D stem cell culture models to study the microcephaly-causing gene WDR62 and identifies a shuttling between Golgi apparatus and mitotic spindle as a possible mechanism of action. Overall the study is well executed and brings new insight into the role of this important gene and its role in progenitor biology.

## Introduction

The development of the human brain is a sophisticated process that extends over several decades, during which a plethora of distinct cell types is generated and assembled into functionally distinct regions and circuits (*Lui et al., 2011*; *Silbereis et al., 2016*). The precise orchestration and coordination of neural stem/progenitor cell proliferation and differentiation are instrumental for the formation

and function of the central nervous system (CNS). Deviations from normal neurodevelopmental processes, such as cell cycle progression, symmetric versus asymmetric cell divisions, lineage commitment, as well as neuronal migration and differentiation, ultimately affect CNS structure and function and can lead to neurological or psychiatric disorders (*Li et al., 2018*).

Mutations in genes involved in the regulation of mitotic progression in neural progenitor cells (NPCs) have been identified in several types of microcephaly (MCPH) (*Thornton and Woods, 2009*; *Phan and Holland, 2021*), a genetically heterogeneous disorder characterized by occipito-frontal circumference 3–4 standard deviations (SD) below the mean of ethnically-, age-, and sex-matched controls (*Woods and Parker, 2013*). Intriguingly, many MCPH-associated proteins are centrosomal or pericentriolar, while others participate in mitotic spindle organization and regulate mitotic progression (*Jayaraman et al., 2018*; *Phan and Holland, 2021*).

Recessive mutations in *WDR62*, located on chromosome 19, are responsible for the second most frequent form of MCPH (*Bilgüvar et al., 2010*, *Nicholas et al., 2010*, *Yu et al., 2010*). *WDR62* encodes a member of an ancient large family of WD-repeat containing proteins involved in coordinating multiprotein assemblies, with the WD domains serving as scaffolds for protein interactions (*Shohayeb et al., 2018*). WD-repeat proteins have pleiotropic functions, ranging from signal transduction and transcriptional regulation to cell cycle control and apoptosis (*Smith et al., 1999*; *Li and Roberts, 2001*; *Stirnimann et al., 2010*; *Jain and Pandey, 2018*). WDR62 is a spindle pole-associated protein involved in mitotic progression and NPC maintenance (*Bogoyevitch et al., 2012*; *Novorol et al., 2013*; *Chen et al., 2014*; *Xu et al., 2014*; *Jayaraman et al., 2016*; *Sgourdou et al., 2017*). In addition, it is thought to play an important role in the assembly and disassembly of the primary cilium (*Zhang et al., 2019*; *Shohayeb et al., 2020*), an antenna-like structure able to sense and convey extracellular cues guiding NPC proliferation versus differentiation (*Wilsch-Bräuninger and Huttner, 2021*). The subcellular localization of WDR62 varies during the cell cycle and is regulated by its N- and/or C-terminal domains and microtubule association (*Bogoyevitch et al., 2012*; *Lim et al., 2015*; *Ramdas Nair et al., 2016*; *Sanchez et al., 2021*). Notably, in interphase, WDR62 is reported to localize to the centrosome via stepwise hierarchical interactions with other MCPH-associated proteins, including CDK5RAP2 and CEP63 (*Kodani et al., 2015*; *O'Neill et al., 2022*). This complex is then stabilized by centriolar satellite proteins and is required for proper centriole duplication (*Kodani et al., 2015*).

We previously reported a homozygous truncating mutation (D955AfsX112) in WDR62 in patients with MCPH and structural brain abnormalities (*Sgourdou et al., 2017*). In the present work, we directly investigated the effects of this mutation on NPC proliferation and neurogenic potential, aiming to gain insight into WDR62 function in corticogenesis. To this end, we generated induced Pluripotent Stem Cell (iPSC) lines from the homozygous patient and heterozygous parent, as well as isogenic corrected iPSC lines obtained via CRISPR/Cas9 gene editing (*Dell'Amico et al., 2021*). From these iPSCs, we then derived and analyzed neuroepithelial stem (iPS-NES) cells, the founders of neural cell complexity (*Onorati et al., 2016*; *Baggiani et al., 2020*), as well as cerebral organoids (COs), cerebrocortical progenitors, and terminally differentiated neurons (*Chambers et al., 2009*; *Shi et al., 2012*; *Arlotta and Paşca, 2019*).

We find that MCPH-associated mutations in WDR62 (D955AfsX112, V1402GfsX12, and W224S) prevent WDR62 localization to the mitotic spindle poles. Moreover, we demonstrate that WT and mutant WDR62 dynamically associate with the Golgi apparatus in iPS-NES cells, COs, and human fetal telencephalic tissue and that the protein shuttles from this organelle to the spindle poles in a microtubule-dependent manner during the interphase-to-mitosis transition. Nocodazole treatment of isogenic control iPS-NES cells prevents WDR62 shuttling from the Golgi apparatus to the spindle poles, mimicking the effects of the genetic mutation. We further demonstrate that the D955AfsX112 mutation impairs mitotic progression and choice of asymmetric versus symmetric divisions in neural progenitors, leading to lineage specification defects and perturbations in the differentiation trajectory of cerebro-cortical neurons derived from mutant iPSCs. Considered together, these findings suggest novel potential mechanisms underlying WDR62 involvement in human neurodevelopment and MCPH etiology.

## Results

### Generation of 2D and 3D in vitro models to investigate WDR62 function in neurodevelopment

We previously reported a consanguineous family with two affected siblings harboring a recessive mutation in *WDR62* (*Sgourdou et al., 2017*). The index case, a 13-year-old male, and his 6-year-old brother presented to medical attention for global developmental delay, severe MCPH, and dysmorphic facial traits. Neuroimaging revealed diffuse pachygyria, thickened cortex, and hypoplastic corpus callosum. Whole exome sequencing identified a homozygous 4 bp deletion in exon 23 of *WDR62*, leading to a frameshift and a premature stop codon, which results in a C-terminally truncated protein (D955AfsX112) (*Figure 1A*). The mutation was confirmed to be homozygous in both affected subjects and heterozygous in both parents by Sanger sequencing (*Sgourdou et al., 2017*). To investigate the consequences of the mutation and gain additional insight into WDR62 function, skin fibroblasts from family members were reprogrammed to iPSCs (parental *WDR62*$^{D955A/WT}$, hereafter 'Het', and patient *WDR62*$^{D955A/D955A}$, hereafter 'Mut') using episomal vectors, assessed for pluripotency, and subsequently differentiated to relevant 2D and 3D neural populations, that is, iPS-NES cells, COs, cerebro-cortical progenitors, and terminally differentiated neurons (*Figure 1B* and *Figure 1—figure supplement 1*). To obtain a gold-standard control for all subsequent analyses, we corrected the mutation through CRISPR/Cas9 gene editing in the Mut iPSC line, generating isogenic (hereafter 'Iso') control lines. We verified sequence restoration in five clones by Sanger sequencing (*Figure 1B*); three (Iso 1, Iso 2, and Iso 3) of which were randomly selected, verified for pluripotency (*Figure 1—figure supplement 1*) and further analyzed.

We examined the expression and localization of WDR62 in the Iso, Het, Mut, and external control (CTRL) iPSC lines by confocal imaging analysis. We found that mutant WDR62 remained diffuse and failed to localize to the mitotic spindle poles in Mut iPSCs (*Figure 1C*). In contrast, in Het and CTRL iPSCs, WDR62 co-localized with PCNT (pericentrin) and TUBA1A (tubulin alpha 1a) at the spindle poles (*Figure 1C* and *Figure 1—figure supplement 2A–C*), similarly to our previous findings in skin fibroblasts (*Sgourdou et al., 2017*). As expected, spindle pole localization was restored in Iso 1 iPSCs (*Figure 1C*). In addition, quantification of WDR62 signal distribution through fluorescence intensity analysis revealed high and narrow fluorescence peaks in Iso 1 and Het iPSCs, similarly to CTRL iPSCs (*Figure 1D*); in contrast, no such peaks were detected in patient-derived Mut iPSCs (*Figure 1C and D*). We also detected WDR62 as a polarized perinuclear signal during interphase, which appeared to be localized around the centrosome (marked by PCTN) in G$_1$-S Mut iPSCs (*Figure 1—figure supplement 2D*).

### WDR62 is associated with the Golgi apparatus in cerebro-cortical progenitors

To explore WT and mutant WDR62 behavior in a neurodevelopmental context, we differentiated Iso 1, Iso 2, and Mut iPSCs toward neural fate by applying a neocortical NES cell derivation protocol (*Morelli et al., 2021*; *Lottini et al., 2022*; *Figure 2A*). Briefly, after the initial induction of dorsal forebrain neuroepithelial identity, based on dual SMAD inhibitors and the WNT inhibitor XAV, we captured long-term self-renewing populations of iPS-NES cells by adding growth factors. iPS-NES cells in culture organize into polarized neural rosettes with the apical domain facing the lumen and express typical neuroprogenitor markers (*Figure 2—figure supplement 1*). Iso 2 and Mut iPS-NES cells expressed comparable levels of *WDR62* mRNA (*Figure 2—figure supplement 2A*) and we were able to detect by western blot the full-length and mutant protein overexpressed in CTRL iPS-NES cells (*Figure 2—figure supplement 2B*).

The pericentrosomal localization of WDR62 in interphase iPSCs (*Figure 1—figure supplement 2D*) prompted us to investigate, by confocal imaging, its potential association with the Golgi apparatus, which is known to be in proximity to the centrosome (*Sütterlin and Colanzi, 2010*). Indeed, WDR62 localized to the Golgi apparatus (GOLGA1/Golgin 97-positive) in interphase Iso 2 and Mut iPS-NES cells (*Figure 2B*). Analysis of fluorescence signal intensity suggested that full-length and mutant WDR62 localized to the Golgi apparatus at comparable amounts, potentially excluding a non-physiological accumulation of the mutant protein (*Figure 2C*). During mitosis, the WDR62 signal was centered, as expected, on discrete regions corresponding to the spindle poles in Iso 2 iPS-NES cells,

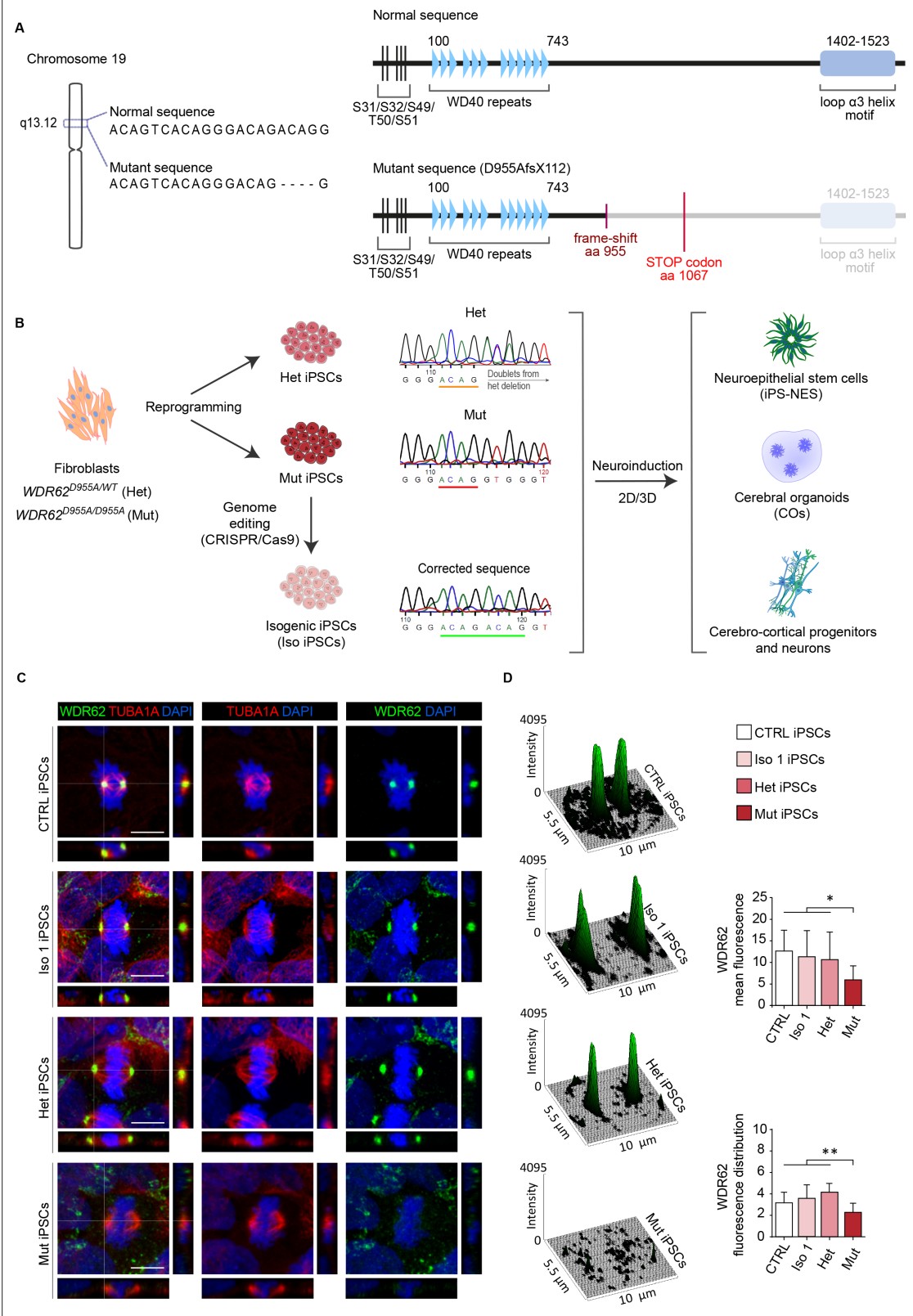

**Figure 1.** Generation of human induced Pluripotent Stem Cell (iPSC)-based 2D and 3D models to analyze WDR62 function in neurodevelopment. (**A**) Schematic illustrations of the *WDR62* gene on chromosome 19q13 indicating the 4 bp (ACAG) deletion in exon 23, and of WDR62 full-length and C-terminally truncated (D955AfsX112) protein structure showing the location of WD40 repeats and loop α3 helix motif. (**B**) Scheme of experimental design. WDR62 iPSCs harboring the heterozygous (Het; D955A/WT) or homozygous (Mut; D955A/D955A) mutation, and isogenic corrected (Iso) iPSC

*Figure 1 continued on next page*

*Figure 1 continued*

lines were employed. Sanger sequencing of iPSCs shows the above-described mutations from one parent and the affected offspring, and the CRISPR/Cas9 corrected sequence. A neuroinduction protocol was applied to Het, Mut, and Iso 1 and/or Iso 2 iPSCs to obtain neuroepithelial stem (iPS-NES) cells, cerebral organoids (COs), as well as cerebro-cortical progenitors and neurons. (**C**) Representative confocal images of WDR62 and tubulin-alpha (TUBA1A) expression in CTRL (external control), Het, Mut, and Iso 1 iPSCs during mitosis. Orthogonal projections indicate WDR62, TUBA1A, and DAPI signals. (**D**) Surface plots and fluorescence intensity analysis show WDR62 signal distribution. Histograms show mean fluorescence intensity (top) and fluorescence signal distribution (skewness, arbitrary unit, bottom) during metaphase. CTRL, Het, and Iso 1 iPSCs show similar WDR62 signal distribution, which is decreased in the measured area in Mut iPSCs (replicates n=3, total cells N=61, p-value <0.05, Kruskal-Wallis test, post hoc Dunn's test in top and bottom histograms. Data are shown as mean ± SD. Scale bar = 10 µm).

The online version of this article includes the following figure supplement(s) for figure 1:

**Figure supplement 1.** Analysis of pluripotency marker expression in patient (homozygous, Mut), parent (heterozygous, Het), and three isogenic-corrected (Iso 1, Iso 2, and Iso 3) induced Pluripotent Stem Cell (iPSC) lines.

**Figure supplement 2.** WDR62 is localized to the spindle poles in CTRL induced Pluripotent Stem Cells (iPSCs) during mitosis.

while the Golgi apparatus was scattered around the chromatin, appearing as clusters formed during mitosis that are eventually partitioned between daughter cells (*Ayala et al., 2020*). In contrast, in Mut iPS-NES cells the WDR62 signal was diffuse (*Figure 2B*), with a large fraction (60%) overlapping with GOLGA1 (*Figure 2C*).

We took several steps to confirm WDR62 localization to the Golgi apparatus. First, we immunostained CTRL iPS-NES cells following siRNA-mediated knockdown of *WDR62*. No signal was detected in *WDR62*-silenced cells (*Figure 2—figure supplement 3A*), indicating antibody specificity. Similarly, in *WDR62* and *LMNA*-Cy3 double-transfected cells, no WDR62 signal was detected in Cy3⁺ cells, thus confirming co-transfection efficiency (*Figure 2—figure supplement 3B*). Conversely, in non-transfected cells, WDR62 signal was detectable both at the spindle poles and at the Golgi apparatus (*Figure 2—figure supplement 3C*), whereas no LMNA signal was detected in Cy3⁺ cells transfected with *LMNA*-Cy3 siGLO siRNA, confirming effectiveness of the transfection control (*Figure 2—figure supplement 3D*, arrowhead). We also induced fragmentation of the Golgi apparatus (*Tie et al., 2016*; *Tie et al., 2017*) in iPS-NES cells by destabilizing microtubules using nocodazole, an agent that interferes with the polymerization of microtubules, thereby leading to the dispersal of functional ministacks throughout the cytoplasm from the canonical pericentrosomal localization (*Cole et al., 1996a*). After a 3 hr treatment (*Figure 2—figure supplement 4A*), ministacks positive for GOLGA1 and GOLGA2 (also known as GM130, a *cis*-Golgi marker) were evident (*Figure 2—figure supplement 4B*). Remarkably, we observed that WDR62 co-localized with GOLGA1 in 58.2%±8.9 of ministacks (*Figure 2—figure supplement 4B*), thus confirming its tight association with the Golgi apparatus.

Next, we co-transfected CTRL iPS-NES cells with FLAG epitope-tagged full-length or mutant (D955AfsX112) WDR62 and GALT1-mWasabi, which localizes to the Golgi apparatus (*Cole et al., 1996b*; *Figure 2—figure supplement 5A, B*), and analyzed transfected cells by confocal imaging. We observed co-localization of FLAG and mWasabi signals, similar to the endogenous WDR62 and Golgi apparatus (GOLGA2⁺) signals (*Figure 2B*), thus corroborating WDR62 presence in the organelle in interphase cells. We extended these observations by testing with the same approach two additional WDR62 pathogenic mutations, N-terminal missense W224S and C-terminal frameshift V1402GfsX12, encountered in patients with MCPH2 (*Bilgüvar et al., 2010*). In both cases, the FLAG signal was absent from the mitotic spindle poles, but detected at the Golgi apparatus (mWasabi and GOLGA2⁺), similar to the pattern observed for the D955AfsX112 mutation (*Figure 2—figure supplement 5C, D*).

To investigate WDR62 function and the consequences of the mutation on cerebro-cortical progenitors in a 3D platform, we derived COs from Iso 2, Het, and Mut iPSCs by culturing in suspension dissociated clumps of iPSCs under cerebro-cortical patterning conditions followed by expansion and differentiation, optimizing a previously published protocol (*Andersen et al., 2020*; *Figure 2—figure supplement 6A*). We performed the analyses at 30 days in vitro (DIV30) (*Figure 2D*), when COs showed an inner structure organized into neural rosettes with SOX2⁺ and nestin⁺ radial glia (RG)-like cells (*Figure 2—figure supplement 6B*). WDR62 localized to the spindle poles of mitotic RG-like cells in Iso 2 and Het, but not Mut, COs (*Figure 2—figure supplement 6C*). A non-canonical and dynamic localization of the Golgi apparatus to the apical process has been reported in bipolar epithelial neural stem cells of the developing mouse telencephalon (*Taverna et al., 2016*). Thus, we asked whether a similar pattern was present in RG-like cells in human COs. We observed a radial distribution of the

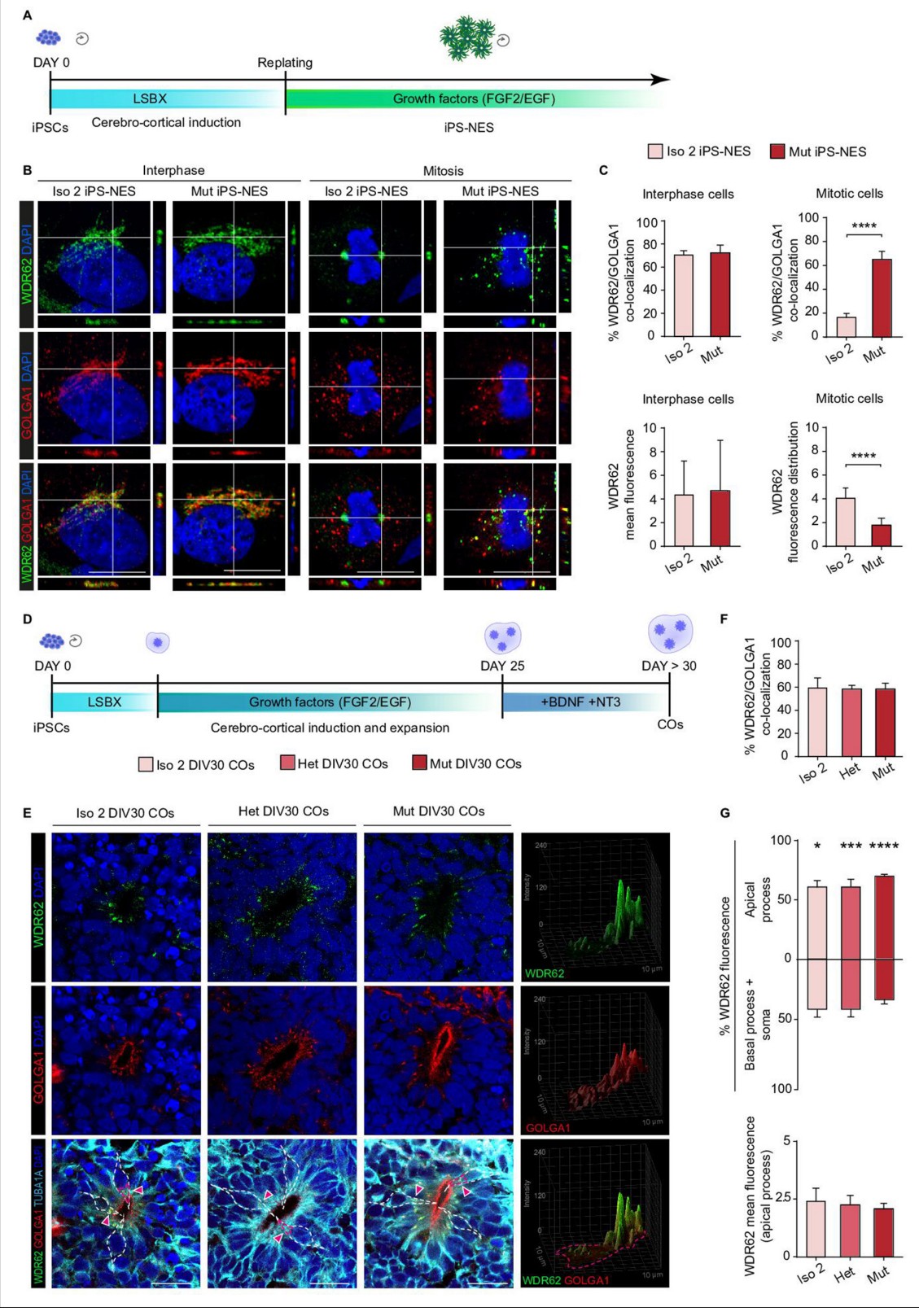

**Figure 2.** WDR62 is associated with the Golgi apparatus in neural progenitors during interphase. (**A**) Schematic illustration of Neuroepithelial Stem Cell derivation protocol from induced Pluripotent Stem Cells (iPSCs) (iPS-NES). (**B**) Immunofluorescence analysis of WDR62 and GOLGA1 in interphase and metaphase iPS-NES cells. During interphase, the WDR62 subcellular localization is comparable in Iso 2 and Mut iPS-NES cells. In metaphase, the WDR62 signal is localized to the spindle poles in Iso 2 iPS-NES cells, but remains diffuse and co-localized with GOLGA1 in Mut iPS-NES cells. Orthogonal

*Figure 2 continued on next page*

*Figure 2 continued*

projections indicate WDR62, GOLGA1, and DAPI signals. (**C**) Quantification of WDR62 and GOLGA1 signal co-localization during interphase (top left histogram) and mitosis (top right histogram); WDR62 mean fluorescence in interphase (bottom left histogram) shows no differences in Iso 2 versus Mut iPS-NES cells. Fluorescence distribution in mitotic cells (bottom right histogram) (skewness, arbitrary units) show differences in Iso 2 versus Mut cells (top left histogram: replicates n=3, total cells N=60, p-value >0.05, unpaired Student's t-test; top right histogram: replicates n=3, total cells N=33, p-value <0.0001, Kolmogorov-Smirnov test; bottom left histogram: replicates n=3, total cells N=164, p-value >0.05, unpaired Student's t-test; bottom right histogram: replicates n=3, total cells N=21, p-value <0.0001, unpaired Student's t-test). (**D**) Schematic illustration of cerebral organoid (CO) derivation protocol from iPSCs. (**E**) Immunofluorescence analysis of WDR62, GOLGA1, and TUBA1A in Iso 2, Het, and Mut COs at 30 days in vitro (DIV30). WDR62 and GOLGA1A signals co-localize to the apical domain of neural progenitors in COs during interphase. 3D surface plots show WDR62 and GOLGA1 signal distribution within the apical process. (**F**) Quantification of WDR62 and GOLGA1 signal co-localization in neural progenitors of Iso 2, Het, and Mut COs at DIV30 (COs n=6, total cell N=193, p-value >0.05, Kruskal-Wallis test, post hoc Dunn's test). (**G**) Quantification of WDR62 fluorescence within the apical process versus the basal process and the soma of neural rosettes of Iso 2, Het, and Mut COs at DIV30 (COs n=6, total cell N=54, p-value <0.05, p-value <0.001, and p-value <0.0001, two-way ANOVA, post hoc Tukey's test top histogram; p-value >0.05, one-way ANOVA, post hoc Tukey's test bottom histogram). Data are shown as mean ± SD. Scale bar = 5 μm in (**B**) and 20 μm (**E**).

The online version of this article includes the following source data and figure supplement(s) for figure 2:

**Figure supplement 1.** iPS-NES cells are polarized progenitors.

**Figure supplement 2.** iPS-NES cells express mutant WDR62.

**Figure supplement 2—source data 1.** Uncropped images of western blots.

**Figure supplement 3.** *WDR62* knockdown supports localization to the Golgi apparatus.

**Figure supplement 4.** Supporting evidence for WDR62 localization to the Golgi apparatus via Golgi fragmentation.

**Figure supplement 5.** Localization pattern of full-length and mutant WDR62 in iPS-NES cells.

**Figure supplement 6.** Characterization of cerebral organoids (COs).

**Figure supplement 7.** WDR62 and GOLGA1 localize to the apical domain of radial glia (RG)-like progenitor cells in cerebral organoids (COs).

**Figure supplement 8.** Full-length and mutant WDR62 overexpression in mouse neural progenitors supports localization to the Golgi apparatus.

Golgi apparatus, polarized toward the ZO-1⁺ apical domain of the RG-like cells within the neural rosettes (*Figure 2—figure supplement 7A*), where WDR62 was also confined (*Figure 2—figure supplement 7B*). We also found that WDR62 and GOLGA1 highly co-localized to the apical domain of the RG-like cells during interphase in Iso 2, Het, and Mut COs (*Figure 2E and F*). Of note, the WDR62 signal was enriched in the apical side of RG-like cells with comparable relative fluorescence values in Iso 2, Het, and Mut COs (*Figure 2G*). Finally, we delivered FLAG epitope-tagged full-length or mutant (D955AfsX112, V1402GfsX12, W224S) WDR62 and GALT1-mWasabi to the developing neocortical wall of the mouse at embryonic day (E) 13.5 by *in utero* electroporation. We observed overlapping FLAG and mWasabi signals in RG processes (*Figure 2—figure supplement 8*). These findings provide evidence of WDR62 localization to the Golgi apparatus in a polarized 3D model of human neural development and suggest that the association of WDR62 with the Golgi apparatus is also observed in the developing mouse neocortex.

## WDR62 shuttling from the Golgi apparatus to the spindle poles is microtubule-dependent

Next, we sought to investigate the mechanism underlying WDR62 shuttling from the Golgi apparatus to the spindle poles at the interphase/mitosis transition. Several MCPH-related proteins have been reported to translocate from various cellular compartments to the spindle poles or centrosomes along microtubular rails (*Wang et al., 2010*; *Welburn and Cheeseman, 2012*). Among them, CDK5RAP2, CEP63, WDR62, and ASPM are known to belong to the same molecular assembly that localizes hierarchically around the centrosome during interphase (*Kodani et al., 2015*).

First, we examined the consequences of microtubule depolymerization on WDR62 localization. Following nocodazole treatment and fixation, WDR62 appeared diffuse around the nucleus in mitotic Iso 2 iPS-NES cells and was absent from the spindle poles and the centrosomal/pericentrosomal area, stained with CETN2 (*Degl'Innocenti et al., 2022*) or PCTN (*Figure 3A*); the same pattern was observed in Mut iPS-NES cells (*Figure 2B and C*). We analyzed WDR62 signal intensity and distribution around chromatin and mean fluorescence values before and after nocodazole treatment (*Figure 3B*), confirming that the observed pattern was comparable to mutant WDR62 distribution in untreated Mut iPS-NES cells (*Figure 2C*). Moreover, the percentage of WDR62 and GOLGA1 co-localization

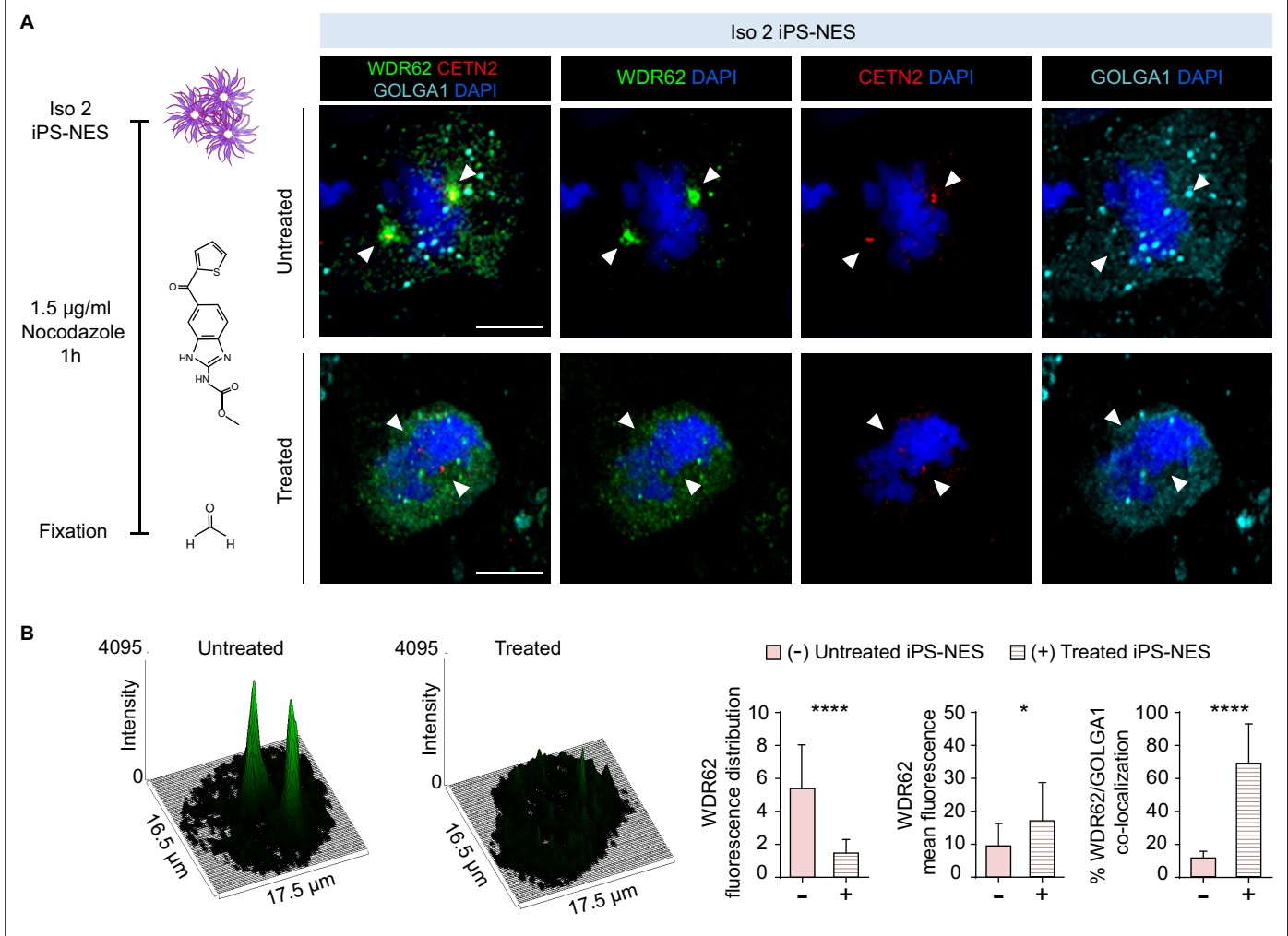

**Figure 3.** WDR62 shuttling from the Golgi apparatus to the mitotic spindle poles depends on microtubules. (**A**) Schematics of the experiment and relative immunofluorescence analysis of WDR62, GOLGA1, and CETN2 in Iso 2 Neuroepithelial Stem cells derived from induced Pluripotent Stem cells (iPS-NES cells) before and after microtubule depolymerization with 1.5 µg/ml nocodazole for 1 hr. In untreated metaphase cells, the WDR62 signal is pericentrosomal. In nocodazole-treated cells, WDR62 remains associated with the Golgi apparatus (marked by GOLGA1). White arrowheads indicate the centrosome (marked by CETN2). (**B**) Surface plots and analysis of WDR62 fluorescence signal distribution in mitotic untreated and nocodazole-treated Iso 2 iPS-NES cells. Histograms show fluorescence signal distribution (skewness, arbitrary units) (left), mean fluorescence intensity (center), and co-localization with GOLGA1 (right) in untreated and nocodazole-treated Iso 2 iPS-NES cells during mitosis. Treated mitotic cells show different WDR62 signal distribution, increased mean fluorescence values, and higher WDR62/GOLGA1 co-localization percentage in the measured area (see also *Figure 2B and C* for comparison with Mut iPS-NES cells). Replicates n=3, total cells N=42, p-value <0.0001, Kolmogorov-Smirnov test, left histogram; replicates n=3, total cells N=42, p-value <0.05, Kolmogorov-Smirnov test, center histogram; replicates n=3, total cells N=32, p-value <0.0001, unpaired Student's t-test, right histogram. Data are shown as mean ± SD. Scale bar = 10 µm.

The online version of this article includes the following source data and figure supplement(s) for figure 3:

**Figure supplement 1.** Impact of the D955AfsX112 mutation on WDR62 interactions with other proteins.

**Figure supplement 1—source data 1.** Uncropped images of western blots.

**Figure supplement 1—source data 2.** Uncropped images of western blots.

**Figure supplement 1—source data 3.** Uncropped images of western blots.

was similar in nocodazole-treated Iso 2 and Mut iPS-NES cells (*Figures 2C and 3B*). Together, these observations strongly suggest that microtubules are fundamental for the Golgi apparatus-to-spindle pole translocation of WDR62 during mitosis and that the C-terminal part of the protein may have an important role in the interaction of WDR62 with microtubules.

We then asked whether the centrosomal localization of CDK5RAP2 was preserved in Mut iPS-NES cells. We found that both during mitosis and interphase, CDK5RAP2 was correctly localized, as we reported previously for skin fibroblasts (*Sgourdou et al., 2017*; *Figure 3—figure supplement 1A*). Next, we investigated CDK5RAP2 localization in Iso 2 iPS-NES cells after nocodazole treatment. We found that, already during $G_1$ and before centriole duplication, CDK5RAP2 was localized to the centrosome, whereas WDR62 remained associated with the Golgi apparatus (*Figure 3—figure supplement 1B*) both in untreated and in nocodazole-treated cells. After centriole duplication at early S-phase (*Holland et al., 2010*), only CDK5RAP2 was detected at the centrosomes both in untreated and in nocodazole-treated cells, even before the centrioles started to migrate to participate in spindle pole formation (*O'Neill et al., 2022*; *Figure 3—figure supplement 1B*). In untreated cells during prometaphase, CDK5RAP2 and WDR62 localized to the centrosome and properly positioned spindle poles, respectively (*Figure 3—figure supplement 1B*). In nocodazole-treated cells, however, the two proteins showed divergent behavior: CDK5RAP2 remained at the centrosome (*Figure 3—figure supplement 1C*), whereas WDR62 appeared dispersed (*Figure 3—figure supplement 1D*).

WDR62 and CDK5RAP2 have been reported to physically interact (*Kodani et al., 2015*). We overexpressed CDK5RAP2 and full-length or D955AfsX112 WDR62 in HEK293T cells and then performed co-immunoprecipitation (co-IP). We found that the mutation did not prevent WDR62 interaction with CDK5RAP2 (*Figure 3—figure supplement 1E*) or with AURKA or TPX2 (*Lim et al., 2015*; *Lim et al., 2016*) in a similar overexpression/co-IP assay (*Figure 3—figure supplement 1F, G*). Together, these findings suggest that the mutation does not impinge WDR62 binding to several of its physical interactors, but prevents shuttling to the spindle poles.

## WDR62 is associated with the Golgi apparatus in human fetal telencephalon

To strengthen our observations of WDR62 localization to the Golgi apparatus in the apical domain of RG-like cells of COs, we examined human fetal telencephalic tissue sections from presumed healthy subjects at 9 and 11 post-conceptional weeks (pcw) (*Figure 4A–D*), when WDR62 is broadly expressed in neural progenitor populations (*Bilgüvar et al., 2010*, *Nicholas et al., 2010*). At 9 pcw in the cortical plate (CP), where committed neuroblasts are populating the incipient cortical layers, the WDR62 signal was polarized and perinuclear, overlapping with GOLGA2, thus retracing the pattern observed in non-mitotic iPS-NES cells (*Figure 4B' and B''*). In the cortical ventricular zone (VZ), where RG cells reside and most mitotic events occur, WDR62 was detected at the TUBA1A-positive spindle poles in mitotic cells (*Figure 4B' and B'''*); in contrast, the WDR62 signal appeared elongated and scattered in interphase cells, reminiscent of the distribution of the Golgi apparatus along radial processes in COs (*Figure 4—figure supplement 1A, A'*). Similarly, in human fetal incipient dorsal and ventral telencephalic areas at 11 pcw (*Figure 4D*), the WDR62 and GOLGA2 signals were polarized and perinuclear in the CP (*Figure 4D and D'* and *Figure 4—figure supplement 1B, B'*) and in the mantle zone of the developing striatum (*Figure 4—figure supplement 1B''*), but scattered and localized in the CTNNB1$^+$ (beta catenin) apical processes in RG cells at the germinative zones (*Onorati et al., 2014*; *Castiglioni et al., 2019*; *Figure 4D' and D''* and *Figure 4—figure supplement 1B', B''*). Thus, we found persistent WDR62 and GOLGA2 signal co-localization at 9 and 11 pcw, showing a perinuclear and polarized pattern at the CP, consistent with the distribution of the Golgi apparatus in post-mitotic neurons (*Taverna and Huttner, 2019*), and a radial/apical pattern at the VZ (*Figure 4F*). Independently from area-specific patterns, WDR62 and GOLGA1 displayed a high incidence of co-localization at the VZ/subventricular zone (SVZ), intermediate zone (IZ)/subplate (SP), and CP (*Figure 4G*).

Taken together, these data suggest a dynamic association between WDR62 and the Golgi apparatus in the developing human telencephalon, supporting the findings in iPS-NES cells and COs.

## WDR62 regulates mitotic progression and asymmetric versus symmetric divisions in cerebro-cortical progenitors

By consensus, MCPH is thought to be a consequence of neural progenitor depletion/premature differentiation (*Jayaraman et al., 2018*; *Phan and Holland, 2021*). For instance, in MCPH3 (caused by

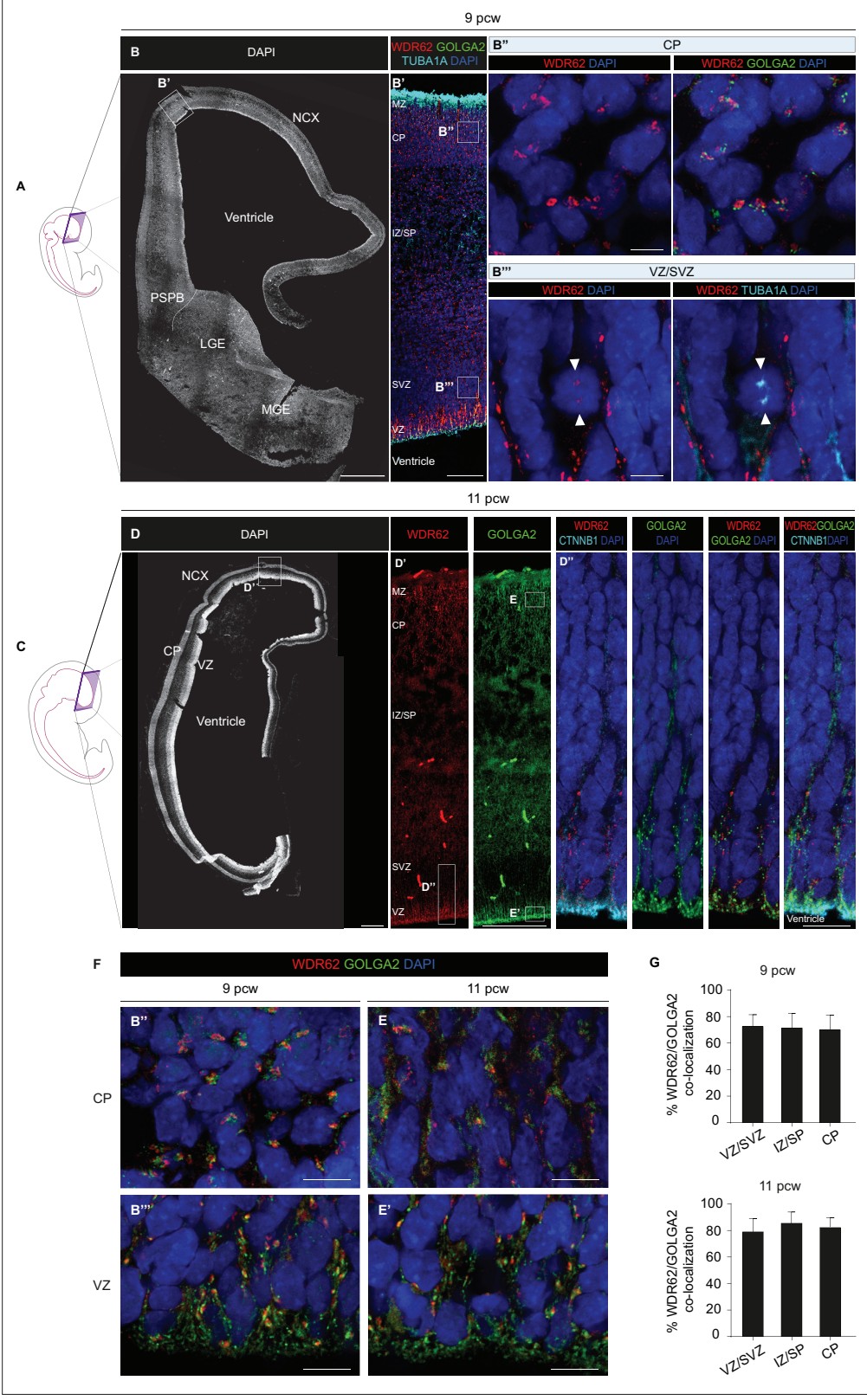

**Figure 4.** WDR62 is associated with the Golgi apparatus in human fetal telencephalic cells. (**A**) Schematic illustration of a human fetal specimen at 9 post-conceptional weeks (pcw) showing the developing central nervous system (CNS). (**B**) DAPI staining of a coronal hemisection of the human telencephalon at 9 pcw. (**B′**) Magnification of the boxed area in (**B**) showing WDR62/GOLGA2/TUBA1A expression in the developing NCX. (**B″**, **B‴**) Magnified views of the boxed areas in (**B′**) showing WDR62/GOLGA2 expression in CP (**B″**) and WDR62/TUBA1A expression in VZ/SVZ (**B‴**). White

*Figure 4 continued on next page*

*Figure 4 continued*

arrowheads indicate WDR62 signal and TUBA1A$^+$ spindle poles in a mitotic cell. (**C**) Schematic illustration of a human fetal specimen at 11 pcw showing the developing CNS. (**D**) DAPI staining of a coronal hemisection of the human telencephalon at 11 pcw. (**D'**) Magnification of the boxed area in (**D**) showing WDR62 and GOLGA2 expression in the developing NCX. (**D''**) Magnified views of the boxed areas in (**D'**) showing WDR62/CTNNB1 and WDR62/GOLGA2 expression at the apical domain of VZ progenitors. (**F**) Representative magnified views of the boxed areas in (**B'**) and (**D'**) showing WDR62 and GOLGA2 signal distribution during interphase at the CP and VZ at 9 and 11 pcw. WDR62 and GOLGA2 signals overlap in post-mitotic neuroblasts at the CP (**B''**, **E**) and in neural progenitor cells (NPCs) at the VZ (**B'''**, **E'**). (**G**) Analysis of WDR62 and GOLGA2 co-localization at the VZ/SVZ, IZ/SP, and CP at 9 and 11 pcw (sections n=2, total Golgi objects N=7087 in 9 pcw sections; sections n=2, total Golgi objects N=5103 in 11 pcw sections). Data are shown as mean ± SD. Scale bar = 500 μm in (**B, D**), 100 μm in (**B', D'**), 5 μm in (**B'', B''', E, and E'**). NCX (neocortex); PSPB (pallial-subpallial boundary); LGE (lateral ganglionic eminence); MGE (medial ganglionic eminence); MZ (marginal zone); CP (cortical plate); IZ/SP (intermediate zone/subplate); SVZ (subventricular zone); VZ (ventricular zone).

The online version of this article includes the following figure supplement(s) for figure 4:

**Figure supplement 1.** Expression pattern and subcellular localization of WDR62 in human fetal telencephalon.

mutations in *CDK5RAP2*) (*Lancaster et al., 2013*), MCPH5 (*ASPM*) (*Li et al., 2017*), and MCPH6 (*CENPJ*) (*Bond et al., 2005*), loss of proliferative capacity of neural progenitors at the VZ/SVZ compromises final brain architecture and size (*Gabriel et al., 2020*; *Phan and Holland, 2021*). For this reason, we sought to determine the consequences of WDR62 dysfunction on iPS-NES cells.

First, we performed phenotypic analyses of Iso and Mut iPS-NES cells by staining with markers of proliferating cells. We found significantly fewer MKI67$^+$ (also known as Ki-67) cells in Mut iPS-NES cell cultures (*Figure 5A*). Since MKI67 detects cells at all phases of the cell cycle except $G_0$, this result suggests a tendency for premature cell cycle exit in Mut iPS-NES cells. On the other hand, we did not find significant differences in the global mitotic index (PH3$^+$/total cells) between Iso 1, Iso 2, and Mut iPS-NES cells in unsynchronized cultures (*Figure 5A*).

Aware that growth factor supplementation can propel iPS-NES cell proliferation, we further investigated cell cycle progression using additional methods. We induced cell cycle arrest through nocodazole (*Figure 5B*) to synchronize iPS-NES cell cultures and evaluated their capability to reorganize cytoskeletal components, such as the mitotic spindle, and proceed through mitosis. Before nocodazole treatment (T0), we did not find statistically significant differences in mitotic figure distribution (i.e. cells at each mitotic phase, evaluated through PH3 staining), between Iso 1, Iso 2, and Mut iPS-NES cells (*Figure 5C*). Nocodazole treatment induced prometaphase arrest and synchronization of the culture after 16 hr (T1) (*Matsui et al., 2012*; *Yiangou et al., 2019*). Following nocodazole removal and 1 hr of release (T2), cells re-enter the cell cycle. We found that approximately 80% of Iso 1 and Iso 2 iPS-NES cells arrested in prometaphase at T1 had re-entered the cell cycle at T2 (*Figure 5C*). In contrast, the percentage of Mut iPS-NES cells in prometaphase was reduced at T1 but increased at T2 (*Figure 5C*). Altogether, our data confirm that after an insult affecting mitotic progression, Mut iPS-NES cells are delayed compared to Iso counterparts.

Previous reports described WDR62 function in regulating mitotic spindle orientation and stability (*Bogoyevitch et al., 2012*; *Miyamoto et al., 2017*; *Guerreiro et al., 2021*) as well as in assembly/disassembly of the primary cilium (*Zhang et al., 2019*; *Shohayeb et al., 2020*), an organelle also involved in cell division choice. Thus, we evaluated inter-centrosomal distance (ICD) and the angle (α) of the spindle pole plane relative to the dish surface (*Figure 5D and E*). We found no differences in ICD, but a smaller spindle angle α in Mut iPS-NES cells (*Figure 5F*). Additionally, we measured length of primary cilia by co-staining with PCTN, which is localized at the basal body during interphase (*Bettencourt-Dias et al., 2011*) and ARL13B, which marks primary cilia (*Caspary et al., 2007*). We found that primary cilia appeared significantly shortened in Mut iPS-NES cells, compared to Iso 2 counterparts (*Figure 5—figure supplement 1*). Taken together, these results suggest a general disruption of cell cycle regulation in Mut iPS-NES cells.

We also quantified symmetric and asymmetric cell divisions in Iso 2, Het, and Mut COs at DIV30 by analyzing the angle (θ) of the cleavage plane relative to the rosette lumen (ZO-1$^+$) – mirroring the VZ cellular composition and organization (*Figure 5G and H*). We found that RG-like progenitors in Mut COs displayed asymmetric cell divisions more frequently, suggestive of premature differentiation (*Kosodo et al., 2004*; *Figure 5I*). On the other hand, the incidence of symmetric and asymmetric

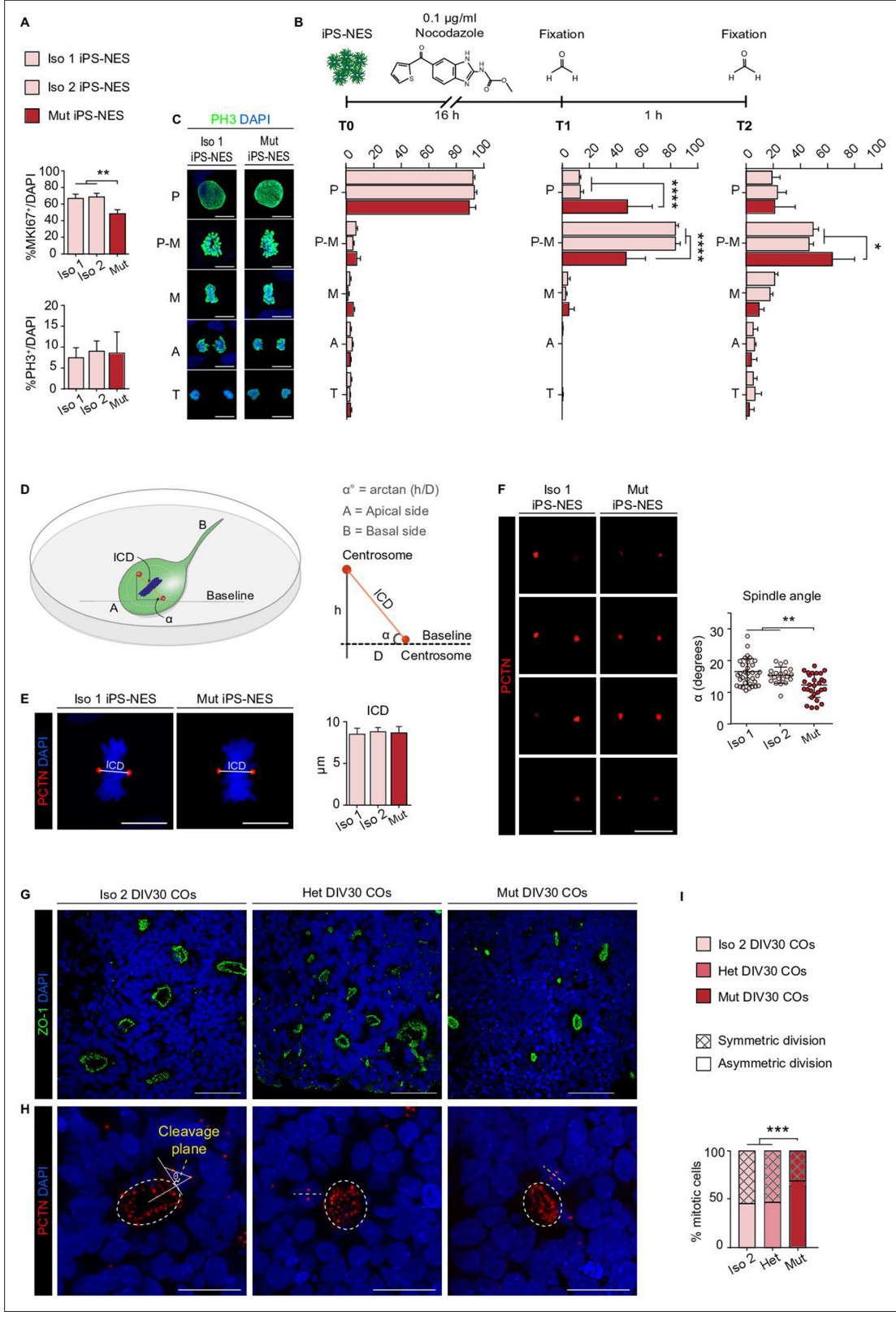

**Figure 5.** WDR62 regulates mitotic progression and asymmetric versus symmetric divisions in neural progenitors. (**A**) Quantitative analysis for MKI67 and PH3 in Iso 1, Iso 2, and Mut Neuroepithelial Stem cells derived from induced Pluripotent Stem cells (iPS-NES cells) reveals a decrease in $G_1$-M (MKI67$^+$) but not in mitotically active (PH3$^+$) cells (MKI67$^+$, replicates n=4, total cell N=3315, one-way ANOVA, post hoc Tukey's test, p-value <0.001; PH3$^+$, replicates n=4, total cell N=12684, one-way ANOVA, post hoc Tukey's test, p-value >0.05). (**B**) Scheme of synchronization experiment in iPS-NES

*Figure 5 continued on next page*

*Figure 5 continued*

cells. (**C**) Representative confocal images and quantification of mitotic figure distribution in iPS-NES cells. The distribution of mitotic figures (quantified as % of mitotic figures at each phase) is not significantly different in unsynchronized (**T0**) Iso 1, Iso 2, and Mut iPS-NES cell cultures (replicates n=4, total cell N=12684, p-value <0.05, two-way ANOVA, post hoc Dunnett's test). Quantitative analysis after 16 hr of treatment (**T1**) shows increased mitotic cell fraction in P-M. The percentage of cells arrested in P-M is higher in Iso 1 and Iso 2 than in Mut iPS-NES cell cultures. Fewer Mut iPS-NES cells proceed to M following nocodazole treatment and subsequent release in standard culture conditions (**T2**) compared to Iso 1 and Iso 2 iPS-NES cells (replicates n=4, total cell N=8764, $P$-value <0.0001, two-way ANOVA, *post hoc* Dunnett's test and replicates n=4, total cell N=9751, p-value <0.05, two-way ANOVA, post hoc Dunnett's test). P (prophase); P-M (prometaphase); M (metaphase); A (anaphase), and T (telophase). (**D**) Schematic 3D representation of inter-centrosomal distance (ICD) and spindle angle α measurements in iPS-NES cell cultures. α was measured as the angle of the spindle plane relative to the surface of the culture dish (baseline). (**E**) Representative confocal images of PCTN and DAPI staining in Iso 1 and Mut iPS-NES cells in metaphase and example of ICD measurements. Quantitative analysis shows no differences in ICD (replicates n=6, total cell N=81, p-value >0.05, one-way ANOVA, post hoc Sidak's test). (**F**) Confocal Z-stack images of PCTN in Iso 1 and Mut iPS-NES cell lines during metaphase. Quantitative analysis shows smaller α in Mut iPS-NES cell compared with Iso 1 and Iso 2 iPS-NES cells (replicates n=6, total cell N=81, p-value <0.01, one-way ANOVA, post hoc Dunnett's test). (**G**) Representative confocal images of ZO-1 in COs of Iso 2, Het, and Mut at 30 days in vitro (DIV30) showing the lumen of neural rosettes. (**H**) Confocal images of PCTN in Iso 2, Het, and Mut COs at DIV30. (**I**) Analysis of the cleavage plane angle $\theta$ in neural progenitors (% mitotic cells) showing a higher percentage of asymmetric (horizontal) divisions in Mut COs compared with Iso 2 and Het COs (COs n=6, total cell N=184, data are shown as %, p-value >0.001, Chi-square test). Data are shown as mean ± SD. Scale bar = 5 μm in (**C**), 10 μm in (**E, F**), and 20 μm in (**G, H**).

The online version of this article includes the following figure supplement(s) for figure 5:

**Figure supplement 1.** Mut iPS-NES cells harbor shorter cilia.

divisions was comparable in Het and Iso 2 COs (*Figure 5I*). These observations suggest that WDR62 is important for maintaining proliferative divisions in RG-like progenitors and that the mutation results in a shift toward neurogenic divisions in COs.

## WDR62 regulates NPC fate

Considering its impact on neural progenitor choice of symmetric versus asymmetric cell divisions, we next investigated potential effects of WDR62 on iPSC differentiation into mature cerebro-cortical neurons and glia. We employed a directed cerebro-cortical differentiation protocol (*Figure 6A*): after an initial induction phase, we obtained, at DIV16, neuroectodermal cells (*Figure 6—figure supplement 1A*), which were subsequently driven toward terminal differentiation through the addition of supplements and neurotrophins (e.g. BDNF) (*Sousa et al., 2017*). Quantification of MKI67⁺ and PH3⁺ mitotic cells, as well as EdU-incorporating S-phase cells, revealed an increase in mitotic index in Mut versus Iso 2 neuroectodermal cells at DIV16 (*Figure 6—figure supplement 1B*).

Next, we investigated the differentiation potential of Iso 2 and Mut neural cells at DIV40 (*Figure 6B and C*) and DIV80 (*Figure 6D and E*). We observed a significant enrichment in early-born neurons expressing TBR1 and BCL11B (also known as CTIP2), markers of layers 6 and 5, respectively, in Mut compared to Iso 2 iPSC-DIV40 cultures (*Figure 6C*). No differences emerged in the percentage of MAP2⁺ cells (*Figure 6C*). However, the total TUBB3⁺ surface area normalized to the total cell number (DAPI⁺), defined as TUBB3 density, was increased in Mut compared with Iso 2 iPSC-DIV40 cultures (*Figure 6C*), suggesting premature differentiation.

Conversely, we observed significantly fewer TBR1⁺ and late-born SATB2⁺ (layer 2/3), but now comparable BCL11B⁺, neurons in Mut compared to Iso 2 iPSC-DIV80 cultures (*Figure 6E*). Overall, the neuronal composition of Iso 2 iPSC progeny at DIV40 and DIV80 was in agreement with previous reports (*Shi et al., 2012*). Moreover, TUBB3 density was no longer different between Mut and Iso 2 iPSC-DIV80 cultures (*Figure 6E*).

By cross-comparing TUBB3 density at DIV40 and DIV80, we observed a significant increase only in Iso 2 cultures (*Figure 6—figure supplement 1C*). Additionally, we measured and cross-compared the average neurite length between Iso 2 and Mut DIV40 and DIV80 cultures. At DIV40 we found, on average, longer TUBB3⁺ neurites in Mut neuroderivatives with respect to their Iso 2 counterparts, while at DIV80 the difference appeared to be resolved, suggesting a possible early neuronal maturation, but not final neuronal morphometric differences between genotypes. By cross-comparing DIV40 and DIV80 time points, we found a significant increase in average neurite length only in Iso 2 cultures (*Figure 6—figure supplement 1C*). Moreover, we found fewer mature neurons expressing the pan-neuronal marker RBFOX3 (also known as NeuN) in Mut compared to Iso 2 iPSC-DIV80 cultures (*Figure 6—figure supplement 1D*), however, both contained terminally differentiated

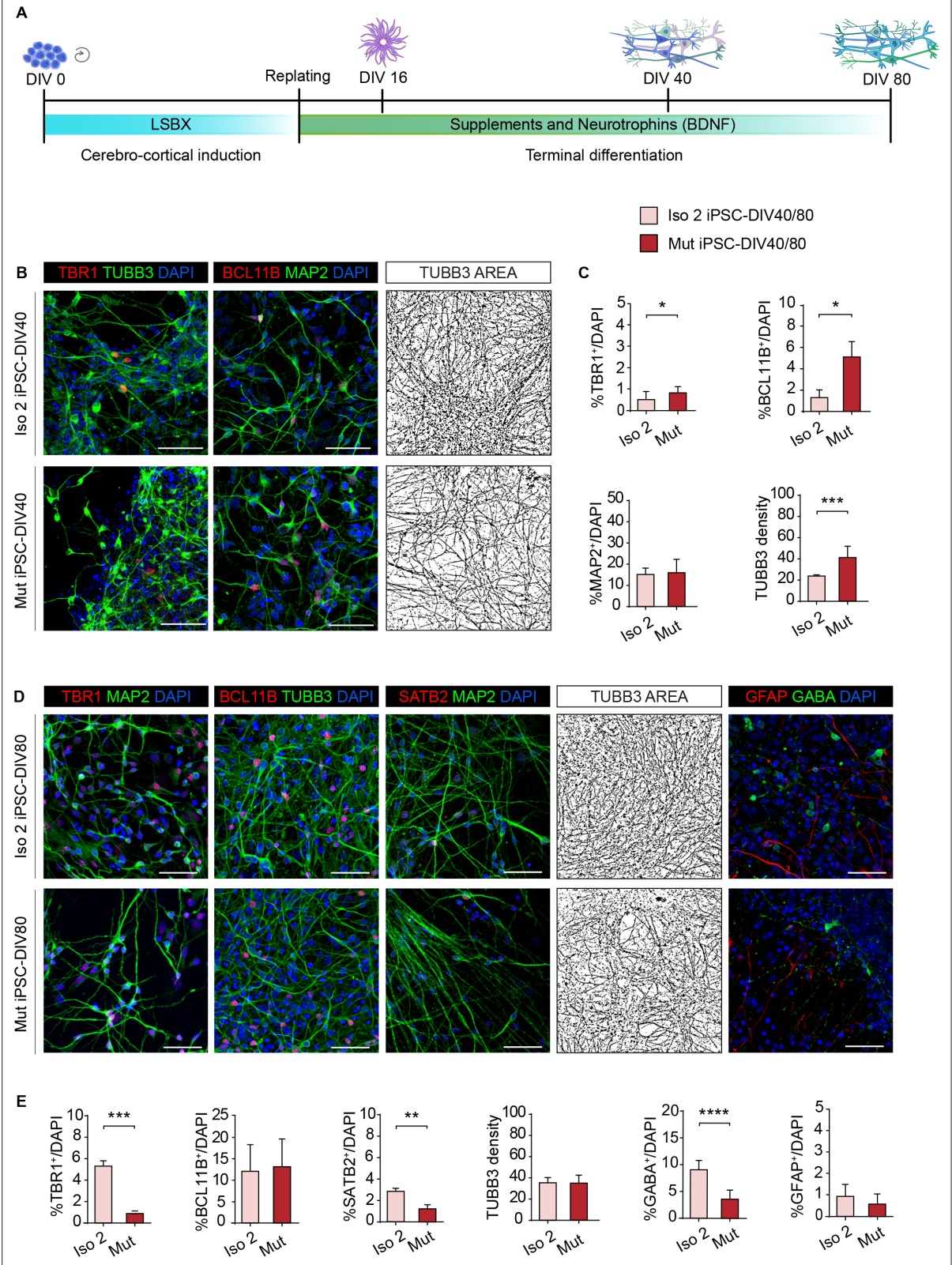

**Figure 6.** WDR62 regulates neural progenitor cell fate. (**A**) Schematic illustration of the cerebro-cortical differentiation protocol (LSBX) via inhibitors of SMAD (LDN193189, SB431542) and WNT (XAV939). (**B**) Representative confocal images of TBR1, BCL11B, MAP2, and TUBB3 in Iso 2 and Mut progeny at 40 days in vitro (DIV40). (**C**) TBR1$^+$ and BCL11B$^+$ neurons are significantly enriched in Mut compared with Iso 2 induced Pluripotent Stem Cell (iPSC)-DIV40 cultures. Quantification shows no differences in MAP2$^+$ cells but significant differences in TUBB3$^+$ density (total TUBB3$^+$ area (μm$^2$)) normalized

*Figure 6 continued on next page*

*Figure 6 continued*

to total cell number (DAPI⁺) between Iso 2 and Mut iPSC-DIV40 cultures (replicates n=3), total cells N=38412, p-value <0.05, unpaired Student's t-test for TBR1; replicates n=3, total cells N=40866, p-value <0.05, unpaired Student's t-test for BCL11B; replicates n=6, total cells N=39595, p-value >0.05, unpaired Student's t-test for MAP2; replicates n=6, total cells N=21013, p-value <0.001, unpaired Student's t-test for TUBB3 density. (**D**) Representative confocal images for TBR1, BCL11B, SATB2, TUBB3, GABA, and GFAP staining in Iso 2 and Mut progeny at DIV80. (**E**) Quantification shows a decrease in TBR1⁺ and SATB2⁺ cells, but no differences in BCL11B⁺ cells and in TUBB3 density in Mut iPSC-derived neurons, and a decrease in GABA⁺, but not in GFAP⁺, cells (replicates n=3, total cells N=23396, p-value <0.05, unpaired Student's t-test for TBR1; replicates n=3, total cells N=29448, p-value >0.05, unpaired Student's t-test for BCL11B; replicates n=3, total cells N=21843, p-value <0.01, unpaired Student's t-test for SATB2; replicates n=3, total cells N=7292, p-value >0.05, Kolmogorov-Smirnov test for TUBB3 density; replicates n=3, total cells N=20375, p-value <0.0001, unpaired Student's t-test for GABA; replicates n=3, total cells N=20370, p-value >0.05, unpaired Student's t-test for GFAP). Data are shown as mean ± SD. Scale bar = 50 µm in (**B**) and (**D**).

The online version of this article includes the following figure supplement(s) for figure 6:

**Figure supplement 1.** Lineage specification defects in Mut induced Pluripotent Stem Cell (iPSC)-derived cerebro-cortical neurons.

neurons expressing the presynaptic marker synaptophysin (*Figure 6—figure supplement 1E*). Lastly, we observed a reduction in GABAergic interneurons, but a normal proportion of GFAP-expressing astrocytes in Mut iPSC-DIV80 derivatives (*Figure 6E*).

Together, these results strongly suggest that WDR62 regulates NPC fate and differentially impacts the neurogenic trajectories of neuronal subclasses, potentially recapitulating the abnormalities in neocortical cytoarchitecture observed in WDR62 mouse model (*Sgourdou et al., 2017*) and MCPH patients.

## Discussion

In this study, we investigated WDR62 function in human cortical development through analyses of a truncating mutation (D955AfsX112) identified in a family with MCPH (*Sgourdou et al., 2017*). We took advantage of patient iPSCs, as well as heterozygous parental and isogenic-corrected counterparts, and iPSC-derived 2D and 3D models of human neurodevelopment: (i) iPS-NES cells, previously employed in mechanistic studies of a rare neuro-ichthyotic syndrome (CEDNIK) (*Morelli et al., 2021*), spinocerebellar ataxia type 3 (Machado-Joseph disease) (*Koch et al., 2011*), Zika virus-induced microcephaly (*Onorati et al., 2016*; *Lottini et al., 2022*), and explored as potential cell therapy (*Dell'Anno et al., 2018*) (ii) COs, also previously applied to MCPH (*Lancaster et al., 2013*; *Li et al., 2017*; *Zhang et al., 2019*), and (iii) cerebro-cortical progenitors and terminally differentiated neurons (*Baggiani et al., 2020*).

Our analyses employing these models demonstrated that WDR62: (i) associates with the Golgi apparatus in interphase neural progenitors and mouse and human fetal telencephalon; (ii) translocates to the spindle poles during the interphase-to-mitosis transition in a microtubule-dependent manner; (iii) regulates mitotic progression and asymmetric versus symmetric divisions in neural progenitors; and (iv) regulates NPC fate.

First, we provide evidence for WDR62 association with the Golgi apparatus during interphase in iPS-NES cells, as previously suggested for HeLa cells (*Nicholas et al., 2010*, *Yu et al., 2010*). We undertook several complementary approaches to establish this association, including immunofluorescence analyses, observation of endogenous WDR62 associating with ministacks following Golgi fragmentation, and of overlapping fluorescence signals after co-expression of WDR62 with established Golgi-localized proteins in iPS-NES cells. We also demonstrate that WDR62 and the Golgi apparatus are both localized to the apical domain of polarized RG-like progenitors in COs, and that WDR62 and Golgi-localized signals overlap in radial processes of mouse neural progenitors. Importantly, we corroborate these observations in tissue specimens of human fetal telencephalon at 9 and 11 pcw, while also describing a radial/apical pattern of WDR62 and the Golgi apparatus in proliferating progenitors in the VZ, and a polarized perinuclear distribution of WDR62 in the CP, consistent with the re-organization of the Golgi apparatus in post-mitotic neurons (*Taverna and Huttner, 2019*). We further demonstrate that translocation of WDR62 from the Golgi apparatus to the spindle poles during the interphase-to-mitosis transition is microtubule-dependent. The truncating mutation we investigated does not disrupt the association of the endogenous mutant protein with the Golgi apparatus in interphase cells or WDR62 interactions with the spindle pole binding partners AURKA

and TPX2, supporting the hypothesis that WDR62 shuttling is microtubule-dependent. Notably, two other MCPH proteins, CDKRAP2 (*Wang et al., 2010*) and CIT (MCPH17) (*Camera et al., 2008*; *Li et al., 2016*), are also associated with the Golgi complex, and several syndromes underlain by genetic defects in Golgi-related proteins present with MCPH (*Dimitrov et al., 2009*; *Banne et al., 2013*; *Sakamoto et al., 2021*; *Fasano et al., 2022*).

Second, we demonstrate that endogenous mutant WDR62 fails to localize to the spindle poles during mitosis and that correcting the mutation with gene editing restores translocation to the spindle poles, thereby causally associating the mutation with the phenotype. Spindle pole localization was previously shown to require AURKA-mediated phosphorylation of the WDR62 N-terminal half (aa 1–790) (*Lim et al., 2016*; *Huang et al., 2021*). However, multiple mutations throughout the protein, including at the C-terminus, have been found to impair localization of endogenous (*Farag et al., 2013*) or overexpressed (*Nicholas et al., 2010*, *Lim et al., 2015*; *Sgourdou et al., 2017*) mutant WDR62 to the spindle poles. Mapping studies have suggested the WDR62-AURKA interaction to require either the WD-repeat domain at the N-terminus (*Huang et al., 2021*) or a middle region (aa 621–1138) (*Chen et al., 2014*) of the protein. The D955AfsX112 mutation is expected to disrupt the latter, as well as the JNK1 phosphorylation site (T1053), previously reported to diminish WDR62 association with the microtubules (*Lim et al., 2015*); however, the interaction of the mutant protein with AURKA appears preserved, at least in a heterologous over-expression system.

As pharmacological disruption of microtubules in isogenic corrected iPS-NES cells prevents WDR62 from translocating to the spindle poles, we are prompted to propose that the C-terminal half (aa 955–1523) of the protein likely regulates WDR62-microtubule interactions required for translocation. Indeed, although microtubule binding is thought to be mediated by the N-terminal half (aa 1–841) (*Lim et al., 2015*), our work suggests that the C-terminal half is also needed, even if the mechanism remains unknown. This is also in agreement with the observation that MCPH-associated C-terminal mutations prevent WDR62 localization to spindle microtubules (*Lim et al., 2015*). We found that two additional pathogenic mutations (W224S and V1402GfsX12) also disrupt spindle pole localization when mis-expressed in iPS-NES cells. Together, these results suggest that both the N- and C-terminal halves of WDR62 play important roles, respectively, binding to and regulating interactions with microtubules. It is worth noting that the Golgi apparatus regulates mitotic spindle formation by promoting the activation of TPX2 (*Wei et al., 2015*), a spindle assembly factor that has been reported to interact with the middle region of WDR62 (*Chen et al., 2014*); this interaction, however, does not appear to be disrupted by the D955AfsX112 mutation.

Third, we demonstrate that WDR62 regulates mitotic progression and the choice of asymmetric versus symmetric divisions in neural progenitors, reminiscent of disrupted centrosome asymmetry we observed in the *Wdr62* gene-trap mouse (*Sgourdou et al., 2017*). We describe a general impairment of the mitotic cycle and failure of cell cycle re-entry in Mut iPS-NES cells, which is aggravated upon cell cycle arrest, consistent with observations in patient fibroblasts (*Sgourdou et al., 2017*). The altered orientation of the mitotic spindle relative to the baseline (i.e. horizontal surface of the dish) and shorter primary cilia in Mut iPS-NES cells, together with the increased incidence of asymmetric cell divisions in RG-like cells in Mut COs, further support the role of WDR62 in mitosis, balance of NPC proliferation and differentiation, and, by extension, neocorticogenesis.

Fourth, we reveal an important role of WDR62 in NPC fate. We document several abnormalities in Mut iPSC-derived cultures undergoing directed cerebro-cortical differentiation, including an enrichment in specific neuronal subclasses (BCL11B$^+$ and TBR1$^+$) at DIV40, and altered neurogenic trajectories at DIV80, with fewer deep- (TBR1$^+$) and upper- (SATB2$^+$) layer neurons, and fewer GABA$^+$ neurons, but no longer overabundant BCL11B$^+$ neurons and apparently normal gliogenesis at DIV80. Consistent with these observations, the Mut iPS-DIV80 cultures contained fewer mature (RBFOX3$^+$) neurons. Together, these findings indicate that the D955AfsX112 mutation perturbs neuronal cell fate commitment secondary to neural progenitor defects, in line with other reports (*Sgourdou et al., 2017*; *Zhang et al., 2019*). Of note, precocious early progenitor delamination and differential effects in early versus late neural progenitor self-renewal and differentiation have been described in *Wdr62* gene-trap mice (*Jayaraman et al., 2016*; *Sgourdou et al., 2017*).

It has been recently reported (*Vargas-Hurtado et al., 2019*) that at early neurogenic stages, mitotic spindles display more astral microtubules and fewer spindle microtubules, whereas the opposite was

observed at later stages. As also suggested by others (*Huang et al., 2021*), it is reasonable to speculate that during the transition from early to late neurogenesis, WDR62 might be involved in the switch in spindle morphology. Thus, WDR62 dysfunction might impact neurogenic trajectories differentially in the course of neurogenesis, eventually altering the final neuronal output observed in MCPH associated with *WDR62* mutations, as also suggested by the selective effect in late progenitors we observed in the *Wdr62* gene-trap mouse (*Sgourdou et al., 2017*).

Considered together, these observations suggest a dynamic and microtubule-dependent shuttling of WDR62 from the Golgi apparatus to the spindle poles. Several pathogenic mutations prevent this process and, therefore, WDR62 localization at the spindle poles during mitosis, thus impacting, in turn, NPC fate and regulation of neurogenic trajectories. Our findings further reinforce the notion that cell cycle impairment and propensity for asymmetric cell divisions in NPCs underscore altered neurogenic processes, ultimately resulting in MCPH (*Jayaraman et al., 2018*). Our results demonstrate that in vitro models of human neurodevelopment derived from patient cells harboring pathogenic mutations are particularly valuable for mechanistic understanding of brain development and pathology, pointing to promising avenues of investigation with broad applications to medicine.

## Materials and methods

All cell and tissue work was performed according to the National Institutes of Health guidelines for the acquisition and distribution of human tissue for bio-medical research purposes and with approval by the Human Investigation Committee and institutional ethics committees of each institution that provided samples. Appropriate informed consent was obtained, and all available non-identifying information was recorded for each specimen.

The frozen sections of human fetal brain used in this study were previously obtained by *Bocchi et al., 2021*, and derived from post-mortem specimens as approved by the University of Cambridge and Addenbrooke's Hospital in Cambridge (protocol 96/85).

All mouse work was according to a protocol approved by the Institutional Animal Care and Use Committee (IACUC protocol #2022-10886) of Yale University.

### Description of the mutation

The mutation under study was identified in index case NG1406-1 and described in *Sgourdou et al., 2017*. Briefly, analysis of whole exome sequencing data detected a novel homozygous 4 bp deletion in exon 23 of *WDR62*, leading to a frameshift and the generation of a premature stop codon at position 1067 producing a C-terminally truncated protein (D955AfsX112). The parents of the affected siblings were both found to be heterozygous for the mutation. iPSCs derived from skin fibroblasts of the index case NG1406-1 and from one parent (NG1406-4) have been employed (*Sgourdou et al., 2017*).

### iPSC culture and maintenance

Reprogramming of patient and parent-derived fibroblasts (NG1406-1 and NG1406-4) was performed via episomal vectors – pCXLEhOCT3/4-shp53-F, pCXLE-hSK, pCXLE-hUL (*Sgourdou et al., 2017*). Human iPSCs were cultured as previously described (*Sousa et al., 2017*). Briefly, cells were seeded on Matrigel-coated culture plates (Cat: #356234, Corning, 1:60) and maintained in StemFlex Basal medium (Thermo Fisher Scientific; #A3349201) or Essential 8 Medium (Thermo Fisher Scientific, #A2858501). Cells were typically passaged with EDTA (0.5 mM) at room temperature (RT). After 3–5 min of incubation, the EDTA solution was removed and cells were gently detached from the dish with a small volume of medium, generating clumps of six to eight cells. In standard conditions (37°C, 5% $CO_2$), iPSC colonies typically grow within 4–5 days.

### CRISPR/Cas9 gene editing: isogenic line generation

CRISPR/Cas9 technology was used to generate isogenic correct lines (*Skarnes et al., 2019*). $1.5 \times 10^6$ WDR62 iPSCs (clone 1b2-1) were nucleofected with pre-mixed 8 µg of synthetic chemically modified sgRNA_1 (Synthego), 200 pmol/µl of single strand donor oligonucleotide (ssODN; IDT) together with

20 µg of HiFi Cas9 Nuclease V3 (Cat: #1081060, IDT) using Amaxa 4D-Nucleofector (Cat: #AAF-1002B, Lonza) transfection system with Primary Cell Optimization 4D-Nucleofector X Kit (Cat: #V4XP-9096, Lonza) in P2 buffer – program DS-150.

After nucleofection, cells were counted and assessed for viability with trypan blue (cell death around 18%), seeded onto Matrigel-coated culture plates supplemented with 10 µM Y-27632 (Cat: #72308, STEMCELL Technologies) and incubated at 32°C (5% $CO_2$) for 2 days. This passage was instrumental to increase the rate of homologous recombination (*Guo et al., 2018*). The day after nucleofection, cell medium was replaced to remove Y-27632. Two days later, cells were detached and plated at low density in 60 cm$^2$ Petri dishes pre-coated with Matrigel and cultured for 15 days, until cell colonies were ready to be picked. Three clonal iPSC lines Iso 1, Iso 2, and Iso 3 were selected and fully characterized. No interclonal variability was observed in the different assays. The specific Iso iPSC line used for each assay is indicated in each figure.

To assess *WDR62* sequence restoration, single clonal populations were first screened through melting curve analysis (*Erali et al., 2008*) performed on a CFX96-BioRad cycler. WDR62_F1 and WDR62_R1 primers were used in combination and samples were prepared using the SensiFast HRM kit for RT-qPCR-melt curve reaction as follows. Amplification phase: Initial denaturation: 95°C 5 s; denaturation: 95°C 10 s; annealing: 60°C 20 s; extension: 72°C 20 s; cycles number: 40. Melting phase: 95°C 30 s. Melt curve: 65–95°C, increment 0.1°C – 5 s.

The relevant portion of genomic DNA from selected clones was amplified by PCR using WDR62_GF1 and WDR62_GR1 primers, with the following conditions: Initial denaturation: 95°C 2 min; denaturation: 98°C 20 s; annealing: 55°C 30 s; extension: 72°C 1 min; final extension: 72°C; 5 min. Cycle number: 30. Finally, the genotype was confirmed by Sanger sequencing. Sequences for genome editing (5′–3′): sgRNA_1: GAAGTGACAGTCACAGGGAC; ssODN: GCCAGTGAGCTCATCCTCTACTCTCTGGAGGCAGAAGTGACAGTCACAGGGACAGACAGGTGGGTGTCCTTTCCACAAGGGAGCCTTAGTTGGAGGAACCCCCAGCTG Primer sequences (5′–3′): Melting analysis: F1: CTCTGGAGGCAGAAGTGACAG R1: CTTGGTGGAAAGGACACCCAC. PCR: GF1: ACTGGGTTTCCTATTCTTGAACTTG GR1: AGGACTTCAGCTGGAGACTCAAC. Sequencing: SF1: TGTGCTGTCTTCCCCATAGTC; SR1: CCCATCCAGGCCTCAACTGTC.

## Generation of cerebro-cortical neurons

Mature neurons were generated from iPSCs through an optimized cerebro-cortical differentiation protocol, based on a dual SMAD inhibition strategy (*Chambers et al., 2009*; *Shi et al., 2012*). As previously described (*Sousa et al., 2017*), iPSCs were dissociated into a single-cell suspension with Accutase (Corning, #25–058 CI) pre-warmed at 37°C and plated onto Matrigel-coated six-well plates at high density (0.7–2×10$^5$ cells/cm$^2$) in StemFlex basal media supplemented with Y-27632 (10 µM). When they reached a confluent state, maintenance medium was replaced with neural induction medium 1:1 DMEM/F-12 (Thermo Fisher Scientific, #31330095) and Neurobasal medium (Thermo Fisher Scientific, #21103049) supplemented with N-2 (1:100, Gibco, #17502-048), B-27 (1:50, Thermo Fisher Scientific, #17504-044), 20 µg/ml insulin (Sigma-Aldrich, #I9278-5ML), L-glutamine (1:100, Thermo Fisher Scientific, #25030-081), MEM non-essential amino acids (1:100, Gibco, #11140-050) and 2-mercaptoethanol (1:1000, Thermo Fisher Scientific, #31350010). For cerebro-cortical induction (LSBX), 100 nM of LDN-193189 (STEMCELL Technologies, #72144), 10 µM of SB-431542 (Merck, #616464-5 MG) and 2 µM of XAV939 (STEMCELL Technologies, #72674) were added. Induction medium was replaced daily until DIV11. For terminal differentiation, cells were dissociated at DIV12 with Accutase and replated onto poly-D-lysine (Sigma, #P6407, 10 µg/ml)/laminin (Invitrogen, #23017-015, 3 µg/ml) coated chamber slides at the density of 8×10$^4$ cells in terminal differentiation medium containing Neurobasal supplemented with N-2 (1:100), B-27 (1:50), L-glutamine (1:100), BDNF (30 ng/ml, R&D Systems, #248-DB025), and Y-27632 (10 µM) to increase cell viability. Culture media were partially replaced every 3–4 days until DIV80.

## iPS-NES cell generation

To obtain iPS-NES cells, 2×10$^6$ cells were replated at the end of the neuroinduction phase (DIV12) onto POLFN-coated six-well plates (0.01% of poly-L-ornithine, Sigma, #P4957; 5 µg/ml laminin; 1 µg/ml fibronectin, Corning, #354008) in NES medium containing DMEM/F-12 supplemented with B-27 (1:1000), N-2 (1:100), 20 ng/ml of FGF-2 (Gibco, #13256 029), 20 ng/ml of EGF (Gibco, #PHG0311),

glucose (1.6 g/l), 20 µg/ml insulin, and 5 ng/ml of BDNF. NES cells were then cultured as previously described (*Dell'Anno et al., 2018*); to preserve optimal growth properties, the medium was replaced every 2–3 days and cells were passaged 1:2 or 1:3 weekly with 0.25% Trypsin/EDTA (Thermo Fisher Scientific, #25200-056) pre-warmed at 37°C.

## Generation and characterization of COs

COs were generated according to a previously published procedure (*Andersen et al., 2020*) with modifications. Briefly, Iso 2, Het, and Mut iPSCs were detached into small clumps with 0.5 mM EDTA. Cells suspension was plated onto poly-(2-hydroxyethyl methacrylate) (also known as poly-HEMA) (Sigma, #P3932-10G) coated dished to avoid cell adhesion and StemFlex medium supplemented with Y-27632 (20 µM) to increase cell viability. The following day, once small aggregates formed, the media were replaced with Neural Induction Medium supplemented with LDN-193189 (100 nM), SB-431542 (10 µM), XAV939 (2 µM), and Y-27632 (20 µM). Neural Induction Medium was partially replaced every other day until DIV6 and replaced with Neural Expansion Medium (NEM); Neurobasal supplemented with B-27 (1:50), L-glutamine (1:100), FGF2 (20 ng/ml), and EGF (20 ng/ml) at DIV7. NEM was partially replaced every other day until DIV25. From DIV25 to DIV30, FGF2 and EGF were substituted by BDNF (10 ng/ml) and NT3 (20 ng/ml) in the Neuronal Differentiation Medium. COs were cultured in static conditions until DIV30. At DIV30, COs were fixed in 4% formaldehyde (FA) for 30 min, washed three times with D-PBSX (0.1% vol/vol Triton X-100 in D-PBS 1×), dehydrated in 30% sucrose overnight and embedded in OCT (VWR, #361603e) for cryostat sectioning.

## NES cell synchronization

For efficient synchronization, iPS-NES cells were seeded onto POLFN-coated coverslips in a 48-well plate at a density of $10^5$ cells/cm$^2$. To induce prometaphase arrest (T1), cells were rinsed once with D-PBS (Sigma-Aldrich, #D8537) and cultured in NES medium containing 100 ng/ml nocodazole (MilliporeSigma, #M1404) for 16 hr. Then, cells were fixed with 4% FA for 20 min at 25°C.

For cell cycle re-entry analysis (T2), after 16 hr of treatment, nocodazole was removed and synchronized iPS-NES cells were rinsed once with D-PBS, released in NES medium for 1 hr and then fixed in 4% FA for immunofluorescence analysis. For mitotic index and mitotic figure distribution analysis, cells were classified following PH3 staining.

## Microtubule de-polymerization assay

To induce microtubule de-polymerization, iPS-NES cells were first plated onto POLFN-coated coverslips in a 48-well plate at a density of $10^5$ cells/cm$^2$. The day after, 5 µM nocodazole was added to cell culture medium for 1 hr; the cells were then rinsed once with D-PBS and fixed (4% FA for 20 min at 25°C) for immunofluorescence analysis.

## Mitotic spindle angle and ICD estimation in iPS-NES cells

Confocal images of iPS-NES cells immunostained with the pericentrosomal marker PCTN were acquired with a pre-determined Z-stack step size of 300 nm and elaborated with ImageJ software (*Schneider et al., 2012*) (developed by the National Institute of Health; https://imagej.nih.gov/ij). The mitotic spindle angle α was calculated as follows: α=arctan (h/D), with 'h' as the distance between centrosomes on Y-axis and 'D' as the distance between centrosomes on X-axis, manually traced (edge-to-edge distance of PCTN signal). ICD was calculated as follows: ICD = D/cos (α) (see *Figure 2— figure supplement 1* for reference).

## Estimation of cleavage plane angle in COs

Confocal images from sections of COs immunostained with the centrosomal marker PCTN were acquired with a pre-determined Z-stack step size of 500 nm and elaborated with ImageJ software. Mitotic cells within neural rosettes were selected for cleavage angle Θ estimation. Operatively, cleavage angle Θ was calculated with respect to the tangent at the rosette lumen as Θ=arcos (a/b) with 'a' as the distance of the farthest centrosome from the tangent to the lumen and 'b' as the line of sight the two centrosomes intersecting the tangent. Cleavage planes with 45°<Θ<135° have been

considered as 'vertical/symmetric division', while cleavage planes with 0°<Θ<45° and 135°<Θ<180° have been considered 'horizontal/asymmetric division'.

## Measurement of primary cilium length

Serial optical Z-stacks with a step size of 300 nm were used to create Z-projections (maximum intensity projection) of primary cilia for a minimum of 50 cells per genotype. Single cilium length was measured using 'NeuronJ' plugin of ImageJ (*Meijering et al., 2004*). Briefly, after selecting the first pixel mid-point at the cilium perinuclear edge, short subsequent segments were traced manually following cilium shape until the last pixel mid-point. Cilia were measured regardless of cell cycle phase, excluding mitosis.

## Quantitative real-time PCR

RT-qPCR was performed to quantify *WDR62* mRNA in Iso 2 and Mut iPS-NES cells cultures. iPS-NES cells were plated onto six-well plates and lysates were collected in RNAprotect Cell Reagent (QIAGEN, #76526) to stabilize the RNA. Subsequently, RNA purification was performed applying RNAeasy Protect Cell Mini Kit (QIAGEN, #74134) and RNase-Free DNase set (QIAGEN, #79254). cDNA was synthesized using GoScript Reverse Transcription System (Promega, #A5001) according to the manufacturer's instructions. PCR was performed using QuantStudio 3 Real-Time PCR System (Applied Biosystems, A28137) with SensiMixSYBRNo-ROX kit (Meridian BIOSCIENCE, #QT650-05). Thermal cycling conditions: denaturation at 95°C for 10 min and 40 cycles of 95°C for 15 s and 60°C for 1 min. Data are expressed as fold change of *WDR62* gene expression relative to *GAPDH* housekeeping gene, according to the $2^{-\Delta\Delta CT}$ method. Three technical replicates were performed for each experiment. qPCR primers (5'–3') qF_WDR62: GGAGGAAGAGTGTGAGCCAG; qR_WDR62: CTTG CCGTTGGTTAGCAGG.

## EdU labeling

The Click-iT EdU Alexa Fluor 594 Imaging kit (Invitrogen, #C10339) was used to perform EdU labeling on iPSC cultures at DIV16, according to the manufacturer's instructions; 3.5 hr after EdU incubation, cells were fixed, imaged with confocal microscopy, and processed using ImageJ. For quantitative analysis, cells with signal intensity above 50% were considered.

## Immunostaining assay

After cell fixation (4% FA for 20 min at 25°C or methanol for 7 min at –20°C for the CDK5RAP2 antibody), three washes with D-PBSX were performed. Then, samples were incubated at RT in permeabilization solution (0.5% vol/vol Triton X-100 in D-PBS $Ca^{2+}/Mg^{2+}$) for 10 min and then in blocking solution (5% FBS, 0.3% vol/vol Triton X-100 in D-PBS $Ca^{2+}/Mg^{2+}$) for at least 1 hr (RT). Subsequently, samples were incubated in antibody solution (3% FBS, 0.2% vol/vol Triton X-100 in D-PBS $Ca^{2+}/Mg^{2+}$) with primary antibodies overnight (ON) at 4°C. The next day, samples were washed three times with D-PBSX before incubation with the corresponding secondary antibodies (all diluted 1:500) and DAPI (Sigma, #32670-25 mg) for 1 hr (RT). Samples were mounted with Aqua-Poly/Mount (VWR, #87001-902) on microscope slides for confocal microscopy.

For immunohistochemistry, human fetal brain sections and COs were first thawed at RT, washed once with D-PBS for 10 min, and permeabilized with 0.5% Triton X-100 in D-PBS for 10 min. Sections were then washed in D-PBS and treated with sodium citrate-based R-buffer A (EMS, #62706-10) in 2100-Retriever (EMS, #62706) at 120°C for 20 min. After antigen retrieval, the sections were washed with D-PBS and blocked with 5% horse serum (Thermo Fisher, #26050070), 1% bovine serum albumin (BSA) in D-PBS with 0.3% Triton X-100 for 1 hr (RT). Primary antibodies were diluted in blocking solution and incubated at 4°C ON. The following day the sections were washed three times in D-PBSX. Secondary antibodies and DAPI were diluted in blocking solution for 1 hr (RT). Then, sections were washed twice in D-PBSX and once in D-PBS and finally mounted with Aqua-Poly/Mount on microscope slides for confocal microscopy. All images were acquired using a laser scanning confocal microscope (Nikon, Eclipse Ti).

## Co-localization analysis

Following immunofluorescent staining, co-localization analysis for WDR62 and GOLGA1 (also known as Golgin97) signals was performed using the DiAna plugin (*Gilles et al., 2017*) (ImageJ) on 300 nm Z-stack confocal images. This plugin allows to calculate co-localization percentage between two selected objects measuring the distance (on three dimensions) of every single pixel unit – defined by size through threshold setting – constituting the object itself. To calculate WDR62-Golgi co-localization, Golgi objects identified by GOLGA1 signal ('object A') were selected as regions of interest (ROI) and the same ROI then identified on the WDR62 channel ('object B'). Then, for each A and B object pair, co-localization of A volume on B was calculated. Alternatively, on human fetal and CO sections, co-localization analysis for WDR62 and GOLGA1 signal was performed using the JACoP plugin (*Bolte and Cordelières, 2006*) (ImageJ) on 300 nm Z-stack confocal images. This plugin allows to calculate percentage of co-localization through Manders' coefficient between two selected objects measuring the co-occurrence (on three dimensions) of every single pixel unit – defined by size through threshold setting – constituting the object itself. To calculate WDR62-GOLGA1 co-localization in CO sections, the apical processes of single RG-like cells were selected as ROI in channel 1 (WDR62) and channel 2 (GOLGA1) composites. Then, co-occurrence of channel 1 and channel 2 pixels was calculated in single RG-like cells.

## Cell culture, transfection, immunoprecipitation, and western blotting

To generate the expression vectors used for co-IP and overexpression experiments, full-length cDNA fragments encoding Myc-hCDK5RAP2, Myc-hAURKA, Myc-hTPX2, or hWDR62-FLAG were amplified by PCR and inserted into a CAG-promoter plasmid backbone by In-Fusion HD (Takara). A plasmid encoding hWDR62-FLAG was generated by amplifying WDR62-FLAG by PCR; the fragment was excised using XhoI and NotI and ligated into a CAG-promoter plasmid. A plasmid encoding WDR62 D955AfsX112-FLAG was generated by overlap extension PCR cloning using two products (aa 1–955 of WDR62 D955AfsX112 and 955–1067 of WDR62 D955AfsX112-FLAG), excised using XhoI and NotI, and ligated into a CAG-promoter plasmid. WDR62 W224S-FLAG was generated by standard PCR mutagenesis. WDR62 V1402GfsX12-FLAG, the mutation was introduced in exon 31 using Gateway with reverse primers targeting the region (for additional details and Gateway oligonucleotides, see *Sgourdou et al., 2017*). The plasmid expressing GALT1-mWasabi was generated by inserting the fragment corresponding to aa 1–61 of hB4GALT1-mWasabi into a CAG-promoter plasmid.

HEK293T cells, maintained in DMEM/10% FBS, were transiently transfected using Lipofectamine 3000 (Invitrogen) according to the manufacturer's instructions. Cell lysates were prepared 24 hr after transfection using lysis buffer (Tris-HCl 50 mM [pH7.4], 1 mM EDTA,150 mM NaCl, 1% NP40, and protease inhibitors). For immunoprecipitation, cell lysates were incubated with FLAG antibody for 1 hr at 4°C, followed by incubation with Dynabeads Protein G (Invitrogen #10003D) ON at 4°C. The immune precipitates were washed three times with cold TBST and analyzed by western blotting.

Overexpression experiments were performed through forward transfection. CTRL iPS-NES cells, maintained in NES medium without antibiotics, were transiently transfected using Lipofectamine 2000 (Invitrogen) according to the manufacturer's instructions. Cells ($0.7 \times 10^5$) were co-transfected with 0.5 µg of WDR62-FLAG (or WDR62 D955AfsX112-FLAG, or WDR62 V1402GfsX12-FLAG, or WDR62 W224S-FLAG) and GALT1-mWasabi in 1:1 molar ratio. Forty-eight hr after transfection cells were fixed with 4% FA at 25°C for 20 min and imaged with confocal microscopy.

$1 \times 10^6$ iPS-NES cells, maintained in NES medium without antibiotics, were transiently transfected using Lipofectamine 2000 as described above. Cells were lysed 48 hr after transfection using lysis buffer (RIPA buffer [Sigma, #R0278] and protease inhibitors [Roche, #11836170001]). After DC assay (Bio-Rad, #5000111) for protein quantification, samples were analyzed by western blotting (anti-WDR62 and anti-TUBA1A antibodies).

## Animals and housing

Animals were housed in groups of four to five in standard caging with *ad libitum* access to water and standard lab chow and cared for in a facility operated by the Yale Animal Resources Center. The facility was maintained on a 12 hr light/dark cycle (lights on at 7 am). Environmental enrichment was provided by nesting material, in accordance with the national standards and recommendations of the Guide for the Care and Use of Laboratory Animals as required by the Public Health Service.

### *In utero* electroporation

All surgeries were performed under sterile conditions and in accordance with an IACUC approved protocol (Yale University). DNA solution (~1 µl) in Hank's Balanced Salt Solution (HBSS, Thermo Fisher Scientific, #88284) containing 0.02% fast green was injected into the embryonic lateral ventricle. After injection, electroporation (four 50 ms square pulses of 30 V and 950 ms intervals for E13.5 embryos) was carried out with forceps-type electrodes (Nepa Gene, #CUY650P3). Final concentrations of the plasmids used were GALT-mWasabi (2 µg/µl), WDR62*-FLAG (0.5 µg/µl; *denotes WDR62-FLAG, WDR62 D955AfsX112-FLAG, WDR62 V1402GfsX12-FLAG, or WDR62 W224S-FLAG). Embryos were collected 24 hr post-IUE.

### *WDR62* siRNA-mediated knockdown

Knockdown experiments were performed through reverse transfection. CTRL iPS-NES cells, maintained in NES medium, were transiently transfected using Lipofectamine RNAiMAX (Thermo Fisher Scientific, #13778150) according to the manufacturer's instructions. Cells ($0.6 \times 10^5$) were alternatively transfected with 100 nM, 200 nM, or 300 nM of siGENOME siRNA-hWDR62 (Dharmacon, #D-031771-01-0002) with similar efficiency (not shown). 100 nM siGENOME siRNA-hWDR62 was selected for immunostaining assay. CTRL iPS-NES cells were also transfected with 100 nM siGLO lamin A/C CTRL siRNA (Dharmacon, #D-001620-02-05) or co-transfected with 300 nm siRNA-hWDR62 and siGLO lamin A/C CTRL siRNA (5:1 molar ratio). Cells were fixed 52 hr after transfection in 4% FA.

### Generation of Golgi ministacks

We followed a previously published methodology to induce Golgi apparatus fragmentation (i.e. Golgi ministacks) (*Tie et al., 2017*). Briefly, $0.7 \times 10^5$ CTRL iPS-NES cells were plated onto POLFN-coated 48-well plate cover glasses. The day after, 10 µg/ml nocodazole was added. After 3 hr, cells were fixed in 4% FA at 25°C for 20 min. For ministack visualization, cells were first rinsed twice with 0.1 mM glycine solution (quenching) and then treated with permeabilization solution (0.5% vol/vol Triton X-100 in D-PBS) for 10 min. Then, cells were alternatively incubated for 1 hr at RT with GOLGA1/GOLGA2 or GOLGA1/WDR62 primary antibodies properly diluted in Fluorescence Dilution Buffer (FDB – 5% (vol/vol) FBS, and 2% (wt/vol) BSA in D-PBS) containing 0.1% saponin. Then, cells were extensively rinsed with D-PBS (3×>10 min washes) and incubated with secondary antibodies diluted in FDB+0.1% saponin for 1 hr at RT. Finally, after 3×>10 min washes in D-PBS, coverslips were mounted with Aqua-Poly/Mount on microscope slides for confocal microscopy.

### Cell lines

All the above-mentioned cell lines (iPSCs, iPS-NES cells, and HEK293T) were routinely tested for mycoplasma.

### Quantitative and statistical analysis

Nuclei and nuclear markers were quantified manually with the 'Cell counter' plugin (ImageJ) or automatically through the 'Analyze particles' function (ImageJ) defining a 15 µm$^2$ particle-size threshold. Mean fluorescence and fluorescence distribution were calculated as mean gray values and skewness, respectively. Operatively, ROIs individuating WDR62 signal were defined with free-hand selection function and measured (ImageJ). Concerning COs, ROIs for analysis corresponding to the cells lining the rosette lumen were identified through either CTNNB1 or TUBA1A signal, which defined clear cytoplasmic boundaries within a few (3–5) Z-stacks. The apical process of the cells was considered as the cytoplasmic portion comprised between the nucleus (stained with DAPI) and the lumen (stained with CTNNB1).

Average neurite length was estimated as the area of TUBB3 signal – automatically determined through the 'Analyze particles' function (ImageJ) – from which the TUBB3 perinuclear signal was removed through the application of a nuclei mask and normalized on the TUBB3$^+$ cell number (quantified manually with the 'Cell counter' plugin [ImageJ]).

Values identify mean and error bars represent SD and a significance level of at least $p<0.05$ (*$p<0.05$; **$p<0.01$; ***$p<0.001$; ****$p<0.0001$). All experiments were performed at least in biological replicates n≥3 or triplicate (rounds of differentiations/batches n=3). The size of the population

(N) is reported for each experiment in the corresponding figure legend. Counts and analyses were performed blinded to conditions and genotypes. Graphs were generated with GraphPad Prism 7.00 software. Images were acquired with a Nikon A1 Confocal Microscope (Eclipse Ti) and NIS-Elements AR 4.20.03 64-bit software. Confocal images were then processed with ImageJ, Adobe Photoshop CC 2019, and Adobe Illustrator 2020.

| Antibody | Brand, Cat. number | Dilution (IF) |
|---|---|---|
| ARL13B | Invitrogen, PA5-61840 | 1:500 |
| BCAT | BD bioscience, BD610153 | 1:150 |
| BCL11B | Abcam, ab18465 | 1:500 |
| CDK5RAP2 | Bethyl Lab, IHC-00063-T | 1:500 |
| CETN2 | Biolegend, W16110A | 1:500 |
| DYKDDDDK [FLAG] | Sigma, F3165 | 1:8000, 1:10,000 WB |
| DYKDDDDK [FLAG] | Invitrogen, MA1-142 | 1: 1000 IP |
| FOXG1 | Abcam, ab18259 | 1:500 |
| GABA | Sigma, A2052 | 1:1500 |
| GFAP | Sigma, g3893 | 1:500 |
| GM 130 (GOLGA2) Alexa Fluor 488 | Abcam, ab275987 | 1:100 |
| Golgin97 (GOLGA1) | Invitrogen, A-21270 | 1:200 |
| LMNA | Cell Signaling, 2025 | 1:200 |
| MAP2 | Millipore, ab5622 | 1:1000 |
| MKI67 | Abcam, ab16667 | 1:300 |
| MYC | Millipore, 06-549 | 1: 5000 WB |
| NANOG | Stemgent, 09-0020 | 1:100 |
| NESTIN | R&D, MAB1259 | 1:200 |
| OCT4 | Stemgent, 09-0023 | 1:100 |
| PCTN | Abcam, ab28144 | 1:1000 |
| PH3 | Millipore, 06-570 | 1:500 |
| RBFOX3 | Millipore, ABN78 | 1:500 |
| SATB2 | Genway, satba4b109 | 1:200 |
| SOX2 | Millipore, ab5603 | 1:400 |
| SSEA4 | Stemgent, 09-0006 | 1:100 |
| SYP | Millipore, MAB329 | 1:500 |
| TBR1 | Abcam, ab183032 | 1:100 |
| TRA-1–60 | Stemgent, 09-0010 | 1:100 |
| TUBA1A | Cell Signaling, 3873 | 1:2000, 1:8000 WB |
| TUBA1A | Bio-Rad, mca77g | 1: 500 |
| TUBB3 | Sigma, T8578 | 1: 500 |
| WDR62 | Bethyl Lab, A301-560A | 1:500 IF, 1:2000 WB |
| ZO-1 | Invitrogen, 617300 | 1:100 |

## Acknowledgements

We thank the patient and the family for tissue donation. We thank Chiara Marchetti, Alessio Zanelli, Catello Guida, Camilla Focacci, and Guglielma De Matienzo (University of Pisa, Department of Biology) for technical support and imaging analysis, Francesco Olimpico (Fondazione Pisana per la Scienza) for assistance in cell culture, and Julia von Blume (Yale School of Medicine, Department of Cell Biology) for advice on generating Golgi ministacks. We thank the reviewers for insightful comments and suggestions.

## Additional information

### Funding

| Funder | Grant reference number | Author |
| --- | --- | --- |
| Ministero della Salute | GR-2018-12367290 | Marco Onorati |
| Wings for Life | WFL-IT-21 | Marco Onorati |
| University of Pisa | PRA_2018_68 | Marco Onorati |
| European Union Next-GenerationEU, National Recovery and ResiliencePlan (NRRP), mission 4, component 2, investiment 1.5 | CUP N. I53C22000780001 | Marco Onorati |
| National Institutes of Health | R01HD075822 | Angeliki Louvi |
| Fondazione Pisana per la Scienza | FPS Grant 2018 | Maria Teresa Dell'Anno |

The funders had no role in study design, data collection and interpretation, or the decision to submit the work for publication.

### Author contributions

Claudia Dell'Amico, Data curation, Formal analysis, Investigation, Writing – original draft; Marilyn M Angulo Salavarria, Yutaka Takeo, Formal analysis, Investigation; Ichiko Saotome, Resources, Methodology; Maria Teresa Dell'Anno, Maura Galimberti, Elena Cattaneo, Resources, Writing – review and editing; Enrica Pellegrino, Resources, Data curation, Investigation; Angeliki Louvi, Marco Onorati, Conceptualization, Supervision, Funding acquisition, Writing – review and editing

### Author ORCIDs

Claudia Dell'Amico http://orcid.org/0000-0002-8234-3423
Marilyn M Angulo Salavarria http://orcid.org/0009-0001-9418-4861
Yutaka Takeo http://orcid.org/0009-0009-8198-2870
Maria Teresa Dell'Anno http://orcid.org/0000-0003-1825-671X
Enrica Pellegrino http://orcid.org/0000-0002-3918-2310
Elena Cattaneo http://orcid.org/0000-0002-0755-4917
Angeliki Louvi http://orcid.org/0000-0003-1737-8707
Marco Onorati http://orcid.org/0000-0001-8517-930X

### Ethics

All cell work was performed according to NIH guide-lines for the acquisition and distribution of human tissue for biomedical research purposes and with approval by the Human Investigation Committee and Institutional Ethics Committee of each institution from which the samples were obtained (University of Pisa Review No. 29/2020 and Yale No. 9406007680). Appropriate informed consent was obtained, and all available non-identifying information was recorded for each specimen. The tissue was handled in accordance with the ethical guidelines and regulations for the research use of human brain tissue set forth by the NIH and the WMA Declaration of Helsinki. The human fetal brain sections used in

this study were previously obtained by Bocchi et al., 2021 (Bocchi et al., 2021) and derived from post-mortem specimens as approved by University of Cambridge and Addenbrooke's Hospital in Cambridge (protocol 96/85).

All mouse work was according to a protocol approved by the Institutional Animal Care and Use Committee (IACUC protocol #2022-10886) of Yale University. Animals were housed in groups of 4-5 in standard caging with ad libitum access to water and standard lab chow and cared for in a facility operated by the Yale Animal Resources Center. The facility was maintained on a 12 h light/dark cycle (lights on at 7 am). Environmental enrichment was provided by nesting material, in accordance with the national standards and recommendations of the Guide for the Care and Use of Laboratory Animals as required by the Public Health Service.

## Decision letter and Author response
Decision letter https://doi.org/10.7554/eLife.81716.sa1
Author response https://doi.org/10.7554/eLife.81716.sa2

---

## Additional files

### Supplementary files
• MDAR checklist

### Data availability
Figure 2_Figure supplement 1 source data 1, Figure 4_Figure supplement 1 source data 2, Figure 4_Figure supplement 1 source data 3 and Figure 4_Figure supplement 1 source data 4 contain the full blots used to generate the figures.

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
