## [Editor Report]

This paper is of interest to neurobiologists studying brain development and potentially to cell biologists in general. The study utilizes 2D and 3D stem cell culture models to study the microcephaly-causing gene WDR62 and identifies a shuttling between Golgi apparatus and mitotic spindle as a possible mechanism of action. Overall the study is well executed and brings new insight into the role of this important gene and its role in progenitor biology.

---

## [Decision Letter]

**Decision letter after peer review:**

Thank you for submitting your article "Microcephaly-associated WDR62 mutations hamper Golgi apparatus-to-spindle pole shuttling in human neural progenitors" for consideration by *eLife*. Your article has been reviewed by 3 peer reviewers, including Anita Bhattacharyya as the Reviewing Editor and Reviewer #1, and the evaluation has been overseen by Marianne Bronner as the Senior Editor.

Essential revisions:

1. This model system lacks polarity, which is important for a correct cell division of radial glia, which in turn is the key process impaired in microcephaly. Assessing the division angle in cells in culture is supposed to model the division angle of radial glia during cortical development. However, within the tissue, radial glia are polarized, with the apical domain facing the ventricle and the ventricular surface can be used as an external reference point to assess the division angle. All this is missing in cells in culture and assessing the division angle to the bottom of the dish is not ideal. Are these cells polarized and is their apical domain facing the bottom of the dish? If authors want to keep this analysis I would suggest that they derive brain organoids from these cells and then measure the division angle within a 3D context, which will be a much more informative and robust way of assessing this.

2. Inherent heterogeneity of the model needs to be better addressed in experiments and results.

a. The data are derived from one patient iPSC line and its isogenic control. This is insufficient for disease modeling studies and the authors need to validate the results in another mutant cell line or through manipulation of the gene in other cells.

b. The results do not indicate whether the total number of neurons (e.g. measured using pan-neuronal nuclear markers) is different between the two genotypes at various DIVs. This is important to in the context of using this system to model microcephaly. The neurons generated in the study were cortical neurons, but there was no evidence showing that they were mature cortical neurons.

c. The statistics, although well defined, are not rigorous. The authors state that the N for some experiments is hundreds or even thousands, but these are technical replicates (n) since the cells are from one person. It's also possible to report N=3 if three rounds/batches of differentiation were used.

3. Limited mechanistic insight

a. There is no direct evidence for a causal relationship between defective mitotic progression and shorter cilia in WDR62 mutant cells.

b. GA localization. The authors should show if WDR62 has a direct interaction with a GA protein by Co-IP (even in a heterologous system).

*Reviewer #1 (Recommendations for the authors):*

The manuscript is well written and the logical progression of experiments is presented well. The use of human patient-derived and isogenic iPSC-derived cells is a strength as is the validation of phenotypes in human tissue. The data shows that loss of function of WDR62 leads to impaired cell cycle progression via altered spindle dynamics that corroborates previous findings in mouse and human cells and cell lines and extends to cells relevant for microcephaly- NPCs and neurons. Despite these strengths, there are substantive concerns that limit the impact of the results and make the manuscript premature for publication in *eLife*.

1. The major shortcoming of the work is the use of cells from only one individual. Validation of the results in other cell lines is required to generalize that this interaction/mutation has effects beyond this one patient. This can be done using cells from another patient or the introduction of the mutation in control cells.

2. Related to #1, the statistics, although well defined, are not rigorous. The authors state that the N for some experiments is hundreds or even thousands, but these are technical replicates (n) since the cells are from one person. It's also possible to report N=3 if three rounds/batches of differentiation were used.

3. While the data corroborates previous data, there is not much new insight into these more relevant cells. More mechanistic insight through manipulation experiments would strengthen the impact of the work.

4. The last piece of data focusing on late progenitors could be expanded, as these progenitors involve reflect human-specific mechanisms.

*Reviewer #2 (Recommendations for the authors):*

1. There are several inconsistencies regarding the employed model systems. In Figure 2 the authors show a reduction in Ki67+ NES cells in mutant cells compared to iso-controls, but no difference in mitosis (PH3+). Instead, a direct differentiation in Figure 5 leads to no difference in Ki67+ and an increase in PH3+ mitoses, further accompanied by very different levels of % of Ki67 cells in the iso-control between the two protocols. This might suggest that the effects could be protocol-dependent and the authors should address this. it would be important for the authors to show with additional markers that their different differentiation protocols work equally efficiently for both genotypes.

2. Assessing the division angle in cells in culture is supposed to model the division angle of radial glia during cortical development. However, within the tissue, radial glia is polarized, with the apical domain facing the ventricle and the ventricular surface can be used as an external reference point to assess the division angle. All this is missing in cells in culture and assessing the division angle to the bottom of the dish is not ideal. Are these cells polarized and is their apical domain facing the bottom of the dish? If authors want to keep this analysis I would suggest that they derive brain organoids from these cells and then measure the division angle within a 3D context, which will be a much more informative and robust way of assessing this.

3. The authors assess the mitotic phases using PH3 staining – high-resolution images of different phases should be shown.

4. GA localization. The authors should show if WDR62 has a direct interaction with a GA protein by Co-IP (even in a heterologous system).

5. Human fetal samples are an important addition. However, the most important site of possible interaction between WDR62 and GA is the apical process of radial glia. Unfortunately, I have found only one supplemental image showing this, but a better resolution should be put in the main figure along with a quantification, as this is highly important in order to corroborate the authors' data from cell culture. Also, labeling the radial processes with a specific marker would also be important to understand if the WDR62 signal in SVZ, IZ, and CP is in the radial processes (note that the basal processes of radial glia are not expected to contain GA) or in other cell types.

6. The authors claim that they do not see evidence of premature differentiation. This is based on MAP2 staining, which is not ideal. They do however see an early increase in layer 5 and 6 neurons and a later reduction in the numbers of upper layer neurons – all potential indications of premature differentiation and depletion of the progenitor pool, that the authors could mention in their Discussion to give a more balanced view.

7. Related to the MAP2 staining, the authors should show if the total number of neurons measured using some of the pan-neuronal nuclear markers is different between the two genotypes at various DIVs. This is important to show also in the context of using this system to model microcephaly.

---

## [Author Response]

Essential revisions:1. This model system lacks polarity, which is important for a correct cell division of radial glia, which in turn is the key process impaired in microcephaly. Assessing the division angle in cells in culture is supposed to model the division angle of radial glia during cortical development. However, within the tissue, radial glia are polarized, with the apical domain facing the ventricle and the ventricular surface can be used as an external reference point to assess the division angle. All this is missing in cells in culture and assessing the division angle to the bottom of the dish is not ideal. Are these cells polarized and is their apical domain facing the bottom of the dish? If authors want to keep this analysis I would suggest that they derive brain organoids from these cells and then measure the division angle within a 3D context, which will be a much more informative and robust way of assessing this.

We thank the Reviewers for this thoughtful comment. We would like to emphasize that NES cells show apico-basal polarity in culture as they are capable of organizing into neural rosettes with the apical domain facing the lumen, more evident especially in the first passages after derivation (new Figure 2 supplement 1). Indeed, several published studies highlight neural rosette formation, strong apico-basal polarization and interkinetic nuclear migration in NES cells (doi: 10.1371/journal.pone.0029597; doi:10.1101/gad.1616208; doi: 10.1523/JNEUROSCI.0130-13.2013; doi: 10.1016/j.celrep.2016.08.038; doi: 10.1038/ncomms7500). The division angle we measured in our analysis is defined by the spindle plane relative to the dish surface. We have clarified this point and included an explanatory schematic in Figure 5.

In addition, in agreement with the suggestion and considering the 3D neural structures as an improvement for our study, we derived cerebral organoids from iPSCs (3 different lines, i.e., homozygous mutant (Mut), isogenic corrected (Iso), and newly included heterozygous (Het); please also see answer to Q2). We analyzed the division angle of radial glia-like progenitors within the neural rosettes, reporting increased incidence of asymmetric (neurogenic) divisions in Mut cerebral organoids at DIV30. Moreover, we investigated in detail the pattern of WDR62 localization in the apical domain of radial glia-like cells of the cerebral organoids (please see also answer to Q9). These data are now presented in Figures 2, 5, and Figure 2 – supplement 6,7.

2. Inherent heterogeneity of the model needs to be better addressed in experiments and results.a. The data are derived from one patient iPSC line and its isogenic control. This is insufficient for disease modeling studies and the authors need to validate the results in another mutant cell line or through manipulation of the gene in other cells.

Respectfully, in our study, we describe a rare de novo *WDR62* mutation reported in a family (NG1406) with MCPH2, which we analyzed and compared to outside CTRL and isogenic retro-mutated iPSC lines and their derivatives. Moreover, we validated our observations by overexpressing WDR62 D955AfsX112 in a heterologous system. In the revised manuscript, we also introduce a Het iPSC line derived from parental fibroblasts of the affected subject, which we tested and compared against control, Iso, and Mut lines for multiple parameters, including generation of cerebral organoids; these analyses are now presented in several figures (Figure 1; Figure 1 – supplement 1; Figure 2; Figure 2 – supplements 6 and 7; and Figure 5).

Furthermore, we analyzed two additional pathogenic mutations reported in patients with MCPH2: N-terminal missense (W224S) and C-terminal truncating (V1402fsX12); doi: 10.1038/nature09327, through expression in iPS-NES cells (Figure 2 – supplement 5). We report that both mutations result in WDR62 delocalization from the spindle poles and retention to the Golgi apparatus during mitosis. Additionally, we introduced *WDR62* full-length and the three mutant expressing plasmids into mouse neocortical progenitors at E13.5 by in utero electroporation. Analyses of E14.5 brains support WDR62 localization to the Golgi apparatus. These data are now included in Figure 2 – supplement 8.

b. The results do not indicate whether the total number of neurons (e.g. measured using pan-neuronal nuclear markers) is different between the two genotypes at various DIVs. This is important to in the context of using this system to model microcephaly. The neurons generated in the study were cortical neurons, but there was no evidence showing that they were mature cortical neurons.

We thank the Reviewers for these comments. We cross-compared the TUBB3-occupied area normalized on total cell number (TUBB3 density) and the average neurite length (neurite TUBB3^+^ area normalized to the number of the TUBB3^+^ cells) at the two time points. We discovered a significant increase in both parameters in Iso 2 iPSC-DIV80 cultures compared with DIV40 counterparts, indicating progressive maturation, while no differences between the two time points were evident in Mut iPSC-derived cultures (Figure 6 – Supplement 1).

On the other hand, both Iso 2 and Mut iPSC-DIV80 cultures contained mature neurons expressing RBFOX3 (NeuN) and Synaptophysin (SYP) (Figure 6 – supplement 1).

c. The statistics, although well defined, are not rigorous. The authors state that the N for some experiments is hundreds or even thousands, but these are technical replicates (n) since the cells are from one person. It's also possible to report N=3 if three rounds/batches of differentiation were used.

Thank you for this comment. In the previous version of the manuscript, we stated in the “Materials and methods” section that all experiments were performed at least in biological triplicates/batches (n ≥ 3) and reported the total size of the population (N) for each experiment in the corresponding figure legend. Of course, even if each line of iPSCs originates from a single subject, during our analyses we employed cells of multiple passages, and performed multiple independent rounds of differentiation, chemical treatments, etc. We now present additional details and specifications regarding statistics in the figure legends, as warranted.

3. Limited mechanistic insighta. There is no direct evidence for a causal relationship between defective mitotic progression and shorter cilia in WDR62 mutant cells.

Several authors have speculated that cilia assembly and disassembly are linked to cell cycle progression. For example, different groups reported that NDE1 is involved in the control of cilium length and influences G1–S progression and that TCTEX1 phosphorylation modulates cilia length and accelerates G1–S in the developing mouse neocortex (doi: 10.1038/ncb2183; doi: 10.1038/ncb0411-340). More specifically, Zhang and colleagues (doi: 10.1038/s41467-019-10497-2) found impaired cilium disassembly in *WDR62* KO organoids, due to disruption of the WDR62-CEP170-KIF2A pathway, whereas Shohayeb and colleagues (doi: 10.1093/hmg/ddz281) reported primary cilia shortening in mouse models harboring 3 different *Wdr62* mutations, proposing, however, a different regulatory pathway involving WDR62-CPAP-IFT88. Our data provide additional information in the general context of WDR62 function in cell cycle progression/cilia integrity. As this is not the primary message of our work, we decided to move the cilia-related data to Figure 5 – supplement 1.

In an attempt to obtain mechanistic insight on WDR62 transport on microtubules, we treated CTRL iPS-NES cells with the kinesin spindle protein inhibitor Ispinesib. This treatment perturbed WDR62 localization, partially impairing shuttling to the spindle poles. We observed that WDR62 failed to localize to the spindle poles in approximately 10% of treated mitotic cells (see Author response image 1). We are happy to share this piece of information with the Reviewers, even though we do not think it adds substantially to our manuscript.

**Author response image 1. sa2fig1:** Kinesin Spindle Protein inhibition partially impairs WDR62 shuttling to the spindle poles. (**A**) Representative confocal images of untreated and Ispinesib (10 nM)-treated CTRL iPS-NES cells. Treated cells were synchronized in prometaphase. (**A’**) Magnification of untreated mitotic iPS-NES cells showing WDR62 localized in the proximity of centrosomes (CETN2 staining). (**A’’**) Magnification of Ispinesib-treated mitotic iPS-NES cells showing divergent WDR62 localization/mis-localization pattern. (**B**) Quantitative analysis of % mitotic cells in untreated and treated conditions and of % mitotic cells with mis-localized WDR62. Mitotic cells were identified through the characteristic nuclear shape indicating chromatin compaction and CETN2 staining, highlighting duplicated mitotic centrosomes. Data are shown as mean ± SD, replicates n=3, total cells N=2085, p-value < 0.0001 (Student’s t-test, *post hoc* Welch’s correction), and %, replicates n=3, total cells N=120, p-value < 0.001 (Fisher’s exact test), in **B** top and bottom histogram, respectively. Scale bars = 20 µm in **A** and 5 µm in **A’** and **A’’**.

b. GA localization. The authors should show if WDR62 has a direct interaction with a GA protein by Co-IP (even in a heterologous system).

We agree that Golgi apparatus localization is a new finding that deserves further support. However, we reasoned that a co-IP experiment might provide misleading outcomes, and a negative result (i.e., lack of direct interaction with one, or more, *bona fide* Golgi apparatus proteins) would be difficult to interpret and would not preclude interactions with *any* Golgi apparatus protein. Moreover, the dynamic and transient localization of WDR62 at the Golgi apparatus might be independent of direct protein-protein interactions. Thus, to address this point, and further corroborate our observations, we induced fragmentation of the Golgi apparatus using nocodazole, thereby enabling formation of Golgi ministacks (doi: 10.1083/jcb.2021091149). We were able to detect WDR62 signal in the ministacks (Figure 2 – supplement 4). Together with other lines of evidence in our study (please also see answer to Q2), this experiment further supports the observation of WDR62 localization to the Golgi apparatus.